

# Towards Gridded Nighttime Aerosol Optical Thickness Retrievals Using VIIRS Day/Night Band Observations Over Regions with Artificial Light Sources

Jianglong Zhang[1], Jeffrey S. Reid[2], Blake Sorenson[1], Steven D. Miller[3], Miguel O. Román[4], Zhuosen Wang[5,6], Robert J. D. Spurr[7], Shawn Jaker[1], Thomas F. Eck[4], and Juli I. Rubin[8]

[1]Department of Atmospheric Sciences, University of North Dakota, Grand Forks, ND, USA
[2]Marine Meteorology Division, US Naval Research Laboratory, Monterey, CA, USA
[3]Cooperative Institute for Research in the Atmosphere, Colorado State University, Fort Collins, CO, USA
[4] Earth Sciences Division, NASA Goddard Space Flight Center, Greenbelt, MD, USA
[5]Earth System Science Interdisciplinary Center, University of Maryland, College Park, MD, US
[6]Terrestrial Information Systems Laboratory, NASA Goddard Space Flight Center, Greenbelt, MD, USA
[7]RT SOLUTIONS Inc., Cambridge MA 02138, USA
[8]Remote Sensing Division, Naval Research Laboratory, Washington DC, USA

*Correspondence to*: Jianglong Zhang (jianglong.zhang@und.edu)

**Abstract.** Using observations from the Visible Infrared Imaging Radiometer Suite (VIIRS) Day-Night Band (DNB), we examined the feasibility of developing a gridded nighttime aerosol optical thickness (AOT) data set based on the spatial derivative of measured top-of-atmosphere attenuated upwelling artificial lights at night (ALAN) over the US, Middle-East and Indian Subcontinent regions for 2017. We also studied the potential of using NASA's standard operational Black Marble nighttime lights product suite (VNP46) for estimating the spatial derivatives of surface artificial light emissions, that is one of the key lower boundary conditions for the retrieval process. Sensitivity of nighttime aerosol retrievals to observing conditions and different methods of estimating the spatial derivative of surface artificial light emissions were also explored. Root-Mean-Square Errors (RMSEs) of ~ ~0.15 and ~0.18 and correlations of ~0.8 and ~0.6 were found between VIIRS nighttime AOT and Aerosol Robotic Network (AERONET) nighttime and daytime data, respectively, suggesting that the proposed gridded nighttime AOT retrievals have reasonable skill levels for potential data assimilation, air quality and climate studies of significant events. We also found that NASA Black Marble products can be used to estimate the spatial derivative of surface artificial light emissions for nighttime AOT retrievals over regions that are not frequently contaminated by aerosol plumes, such as the USA. This study demonstrated the feasibility of constructing a gridded nighttime AOT data, using artificial lights, for monitoring of nighttime aerosol events over large spatial and temporal domains. Given the deployment of VIIRS instruments (currently in orbit and forthcoming) on board the NOAA Joint Polar Satellite System series satellites, this study can be viewed as a precursor for gridded nighttime AOT retrievals at both regional and global scales in the future. We also show that the use of the NASA Black Marble products, which would greatly save processing time of this method, is challenging over regions with frequent aerosol pollution such as the Indian Subcontinent and further exploration is required.



## 1 Introduction

35 The daytime state of atmospheric aerosol particles has been routinely monitored from passive space-borne instruments for decades, through measurement of reflected/scattered sunlight by aerosol layers relative to the surface background. Commonly used instruments include the Advanced Very High Resolution Radiometer (AVHRR; Nagaraja Rao et al., 1989 ), the Moderate Resolution Imaging Spectroradiometer (MODIS; Levy et al., 2013), the Multi-angle Imaging SpectroRadiometer (MISR; Garay et al., 2020), the Visible Infrared Imaging Radiometer Suite (VIIRS; Hsu et al., 2019),

40 and the radiometers included on the Geostationary Operational Environmental Satellite series (GOES; Zhang et al., 2020). However, daytime (i.e., visible-band) sensors and associated retrieval algorithms miss important nighttime aerosol activity, and by extension, the potential for a larger set of observations for aerosol data assimilation and associated improved tracking of major aerosol outbreaks and the better understanding of aerosol diurnal variability and associated effects on air quality, weather and climate.

45 Retrieving nighttime aerosol and cloud properties using passive visible wavelength satellite observations is challenging because the Top-Of-Atmosphere (TOA) upwelling visible light signals at night are significantly weaker and more complex than those of the daytime. Namely, the upwelling nighttime light can come from reflected moonlight or from various terrestrial and atmospheric emission sources such as electric lights and natural gas flaring from anthropogenic activities, and natural light emissions of the night that include wildfires, lightning flashes, aurora, and even some forms of marine

50 bioluminescence (Miller et al., 2013). Daytime-measuring aerosol sensors such as MODIS, MISR, and GOES ABI are simply not designed for the nighttime levels of visible light that would enable aerosol studies; their lower-limit on visible light sensitivity is several orders of magnitude higher than the strongest nocturnal light sources.

 Besides possessing sunlight-sensitive visible and shortwave infrared channels, the VIIRS instrument (carried onboard the Suomi National Polar-orbiting Partnership (Suomi-NPP) and the NOAA JPSS satellite series (e.g. NOAA-20 and NOAA-21)

55 satellites) includes a Day-Night Band (DNB) instrument that is designed to be sensitive and calibrated not only to daytime radiances, but also to nighttime light signals occurring in its broad visible to near-infrared bandpass. These signals include both reflected moonlight and visible light emissions from natural (forest fires) or anthropogenic sources (city lights). Using reflected moonlight, a daytime-like aerosol retrieval method can be directly applied for retrieving aerosol properties at night (e.g. Zhou et al., 2021). The advantage of the moonlight-based method is that if measured TOA radiation from reflected

60 moonlight can be converted to reflectance (Miller et al., 2013), a daytime aerosol retrieval scheme can be directly applied with only a few necessary algorithm changes. This ease of algorithmic portability is because surface reflective properties, as well as the absorbing and scattering characteristics from aerosols and gas molecules, should remain similar for sunlit and moonlit conditions for any given set of wavelength channels. However, this moonlight-based method cannot be implemented for moonless nights, which for a sun-synchronous satellite such as S-NPP or JPSS means that roughly ½ of all

65 nights are not candidates for this method. The lunar availability in fact varies as a function of observation time and location





on Earth, as explored by Miller et al. (2012). Also, over land, complex surface reflectance features, especially for regions with high surface albedos, can pose a challenge for the moonlight-reflectance based AOT retrieval methods.

As an alternative approach, studies have shown that nighttime Aerosol Optical Thickness (AOT) can be retrieved from regions with temporally stable/invariant artificial light sources, either by measuring attenuated artificial light emissions

(Zhang et al., 2008; Johnson et al., 2013) or by detecting changes in the horizontal spatial gradient in the vicinity of selected artificial light sources (e.g. McHardy et al., 2015). Using artificial light emissions from more than 4,000 cities, Zhang et al. (2019) show that AOT can be retrieved on a regional basis with reasonable accuracy. Nevertheless, as noted in those studies, *a priori* knowledge of city/town locations is needed. Also, regions with multiple small towns or very large cities pose spatial representativeness-related issues to these algorithms. Importantly, the natural spatio-temporal variability of these surface

light sources, tied to many factors including human activity, seasonal effects on surface/vegetation properties, and angular-dependent obstructions, all factor into the inherent variance—impacting the uncertainty of any atmospheric composition retrieval that is predicated on the stability of that surface source.

In this study, we aimed to address the issues mentioned above, and eliminate the requirement for prior knowledge about city/town locations. Our approach involved investigating the idea of conducting nighttime aerosol retrievals within equally

sized grid areas across a given region by taking spatial derivative of measured top-of-atmosphere attenuated upwelling artificial lights at night (ALAN). We recognize that some grid cells may not include artificial light sources, while other grid cells containing small settlements (partial filling), and megacities may in fact be divided into multiple grids. We hypothesize that any bias related to spatial representativeness can be suppressed by using equal-area grids for aerosol retrievals. In addition, a NASA Black Marble (VNP46) data product has been developed that provides atmospherically corrected, viewing

angle adjusted artificial light surface emission (Román et al., 2018; Wang et al., 2021). It is important to determine whether NASA's Black Marble nighttime lights products from Suomi-NPP (hereby termed, VNP46) can be used as the lower boundary condition to estimate artificial light emissions in aerosol-free and cloud-free skies, as this information is necessary for nighttime aerosol retrievals. If Black Marble has sufficient skill, its use would save an additional processing step needed to perform an ALAN method retrieval.

In this study, data from the Suomi-NPP VIIRS DNB for an arbitrarily chosen year of 2017 over regions of the US, the Middle East, and Indian Subcontinent were used to:

    1) Investigate the feasibility of developing gridded nighttime AOT retrievals using observed TOA nightlight emissions from artificial light sources;

    2) Investigate the feasibility of using NASA VNP46 data as a proxy for the surface light source emissions as needed

95         for nighttime AOT retrievals;

    3) Explore the sensitivity of retrieval-related parameters on the accuracy of nighttime AOT retrievals.

This paper is organized as follows: Section 2 introduces and describes the datasets included in the study. Section 3 discusses theoretical-bases for retrievals and validation/evaluation methods used. Results and limitations of our proposed method are





discussed and analyzed in Section 4, with Section 5 providing an overall summary of this research, as well as an outline of
next steps.

**2 Datasets**

Suomi-NPP VIIRS data over the US, the Middle East and the Indian Subcontinent from 2017 were used for nighttime
aerosol retrievals; these observations covered a wide range of aerosol conditions and underlying city light structures. The
Suomi-NPP NASA Black Marble data were used for estimating surface artificial light source emissions (Román et al., 2018;
Wang et al., 2021). Surface-based lunar photometer AOT retrievals (Berkoff et al., 2011; Schafer et al. 2024) from the
Aerosol Robotic Network (AERONET) were used for evaluating the VIIRS/DNB-retrieved nighttime AOTs. MISR and
MODIS data were also used to spatially cross-check the VIIRS/DNB AOT retrievals. True color Suomi-NPP VIIRS images,
constructed using VIIRS observations at red, blue, and green channels, were obtained from the NASA World View website
(https://worldview.earthdata.nasa.gov/) for qualitative (visual) inter-comparison with VIIRS nighttime retrievals.
Both the Suomi-NPP VIIRS Environmental Data Records (EDRs) and Sensor Data Records (SDR) data were used in the
study. The VIIRS Day Night Band SDR (SVDNB) provided calibrated radiances, and VIIRS Day Night Band SDR
Ellipsoid Geolocation (GDNBO) from provided both geolocation data (including terrain-correction for surface elevation-
based parallax effects) and other ancillary parameters such as sensor/solar/lunar geometries and lunar phase angle. The
VIIRS Cloud Cover Layer EDR (VCCLO) from the suite of VIIRS EDR products was used for cloud clearing of VIIRS
observations.

The VIIRS DNB is a panchromatic channel with the spectral band ranging from 0.5-0.9 μm with a center wavelength around
0.7 μm, and a spatial resolution of ~750 m that is held nearly constant across its entire ~3000 km wide swath due do a
dynamic sub-detector aggregation technique (Schueler et al., 2013). The VIIRS DNB is designed to have a dynamic
detection range from $3 \times 10^{-9}$ W·cm$^{-2}$·sr$^{-1}$ to 0.02 W·cm$^{-2}$·sr$^{-1}$ (Liao et al., 2013), providing the VIIRS DNB sensitivity to
extremely low-levels of visible/near-infrared light at night, including reflected moonlight and emissions from anthropogenic
(e.g. city lights, gas flares from oil rigs), and natural (forest fires, lightning and aurora) sources (Miller et al., 2013). The
VIIRS EDR and SDR data were obtained from the free and publicly accessible NOAA Comprehensive Large Array-Data
Stewardship System (CLASS) website (https://www.aev.class.noaa.gov/saa/products/welcome).

The MISR instrument provides multi-spectral (446, 558, 672 and 866 nm) observations at nine different viewing angles
ranging from forward 70.5° to backward 70.5°. The current version 23 level 2 MISR aerosol products were used in this study
(Garay et al., 2020) for evaluating the DNB AOT retrievals. Included in the level 2 MISR aerosol products are retrieved
AOT at 550 nm at a spatial resolution of 4.4 km with other ancillary information such as geolocation and viewing
geometries. Seasonally-averaged MISR AOT (550 nm) were constructed at a spatial resolution of 0.5 x 0.5°
(Latitude/Longitude) for the December-January-February (DJF), March-April-May (MAM), June-July-August (JJA) and
September-October-November (SON) seasons.





The MODIS instrument, carried onboard both the Terra and Aqua satellites, provides visible, shortwave infrared, and thermal infrared observations at 36 narrow-band spectral channels. True color MODIS images, constructed using MODIS observations at red, blue, and green channels, were also obtained from the NASA World View website (https://worldview.earthdata.nasa.gov/) for visual inter-comparison with VIIRS nighttime retrievals. Here we also used the operational (collection 6.1) level 2.0 Aqua MODIS Dark Target (DT) product of Levy et al. (2013). The Aqua MODIS DT data are available at a spatial resolution of $10 \times 10$ km². Quality assurance steps were applied, including using only retrievals with cloud fraction less than 80%, and allowing only "best" quality over land retrievals, and "marginal" and better quality retrievals over ocean, as indicated by quality flags in the products (Shi et al., 2011).

After quality-checking the MODIS data, seasonal averages of MODIS DT AOT (550 nm) were constructed at a spatial resolution of 0.5 x 0.5° (Latitude/Longitude), similar to the MISR gridded retrievals. Over highly reflective (at visible/near-infrared wavelengths) surfaces such as desert regions, no MODIS DT retrievals are available. Here, we enlisted the operational Aqua MODIS Deep Blue (DB; level 2.0, collection 6.1) AOT retrieval product of Hsu et al. (2013). By using MODIS observations from blue wavelengths, where which surfaces are relatively less reflective than in green or red channels, aerosol retrievals can be performed over the desert regions. Similarly, seasonal averages of MODIS DB AOT (550 nm) were constructed at a spatial resolution of 0.5 x 0.5° (Latitude/Longitude).

As one of the NASA Black Marble data products (VNP46), the VIIRS Lunar BRDF-Adjusted Nighttime Lights Monthly L3 Global 15 arc-second linear Latitude/Longitude grid data provide cloud-free, atmosphere and lunar BRDF effect corrected surface nighttime light emissions (~500m at the equator). We used the NASA VNP46 data to check the potential and possible issues of using the surface nighttime artificial light emissions as estimated by the VIIRS VNP46 data for nighttime aerosol retrievals.

To evaluate the performance of nighttime aerosol retrievals, the newly available version 3, level 1.5 lunar AERONET data were used. The lunar AERONET AOT data at 440, 675, 870, 1020 and 1640 nm were derived by measuring attenuated moonlight between waxing and waning quarter moons (Berkoff et al., 2011; Schafer et al. 2024). AERONET utilizes the Robotic Lunar Observatory (ROLO) model of lunar irradiance (Stone and Kiefer, 2004) with AERONET empirical corrections as a function of lunar phase angle determined from lunar Langley calibrations of Cimel sun photometer instruments at high altitude observatories (Mauna Loa and Izaña). The cloud screening and quality assurance algorithms for lunar AOD are the same as applied to the daytime solar AOD data (Giles et al., 2019) however the aureole curvature check for cirrus was not possible due to insufficient lunar aureole intensity. To inter-compare with VIIRS/DNB AOT retrievals centered at 700 nm, the lunar AERONET AOT data at 675 nm were used in this study. The spatial and temporal collocation windows for nighttime AERONET and VIIRS/DNB AOT data are ±0.3° (Latitude/Longitude) and ±30 minutes respectively. Note that lunar AERONET data are available only over moon-lit nights when the sensor's line-of-sight to the lunar disk is not obscured by clouds. Thus, we also used the Version 3, level 2.0, quality assured daytime AERONET data (Giles et al., 2019), which has an AOT uncertainty on the order of 0.01-0.02 with the higher errors in the UV channels (Eck et al., 1999). Daytime AERONET data can provide some indication of the performance of satellite retrieved nighttime AOTs even at



nights without moonlight. The spatial and temporal collocation windows for daytime AERONET and nighttime VIIRS AOT data are ±0.4° (Latitude/Longitude) and ±24 hours respectively. We caveat that diurnal variations in AOT values can be nontrivial. Thus, for a given night, to inter-compare nighttime VIIRS data, we require the difference between the averaged daytime AERONET data for the day before and the day after a given night to be less than 0.2 (675 nm) to avoid days/nights with large variations in AOT values. This assumption of pseudo-persistence can overcome most uncertainty, with the

exception of a nocturnal transient in the AOT.

## 3       Methodology

### 3.1 Nighttime aerosol retrieval method

Our nighttime AOT retrieval method follows Zhang et al., (2019). In that study, the spatial variation in TOA radiances for a given region of artificial light sources, assumed to be the standard deviation (STD) for that source region, is first estimated

over pre-selected cloud-free and likely aerosol-free (or to be precise, low aerosol loading, since some amount of aerosol is always present) conditions. With the presence of an aerosol layer, the spatial variation in TOA radiances for the artificial light source is expected to be reduced relative to the background low aerosol loading and cloud-free condition due to multiple scattering effects and 'smearing' of highly varying structural details. That is, under clearer conditions a nighttime scene shows a stronger contrast to those that have a heavy aerosol particle loading. Thus, the observable changes in spatial

variation in VIIRS/DNB-measured TOA radiances for a given artificial light source contain aerosol information useful for nighttime AOT retrievals.

The concept of spatial variation/contrast suppression is illustrated by Figure 1. Figure 1a shows the nighttime VIIRS/DNB image over the intersection of 3 USA States (Wyoming, Nebraska and Colorado) at 08:06 UTC on September 03, 2017. On this day, no visible aerosol plume was observed over the region, as also suggested from the VIIRS true color image obtained

at 19:30 UTC on September 03, 2017 (see Figure 1b). Just one day later, this same region is covered by optically thick smoke plumes, as shown in Figures 1c (nighttime VIIRS/DNB image) and 1d (daytime VIIRS true color image). Comparing Figure 1a (aerosol free) and 1c (aerosol polluted), we see that background regions are much brighter due to the reflection of moonlight (the Moon was in waxing gibbous phase during this period) by the aerosol layer. Also, both the intensity and the spatial contrast reduction (i.e., blurring effect) of artificial light sources are reduced in the presence of aerosol plumes. The

reduction in intensity and spatial variation of artificial light sources can both be linked to the optical thickness of the intervening aerosol layer (Zhang et al., 2019).

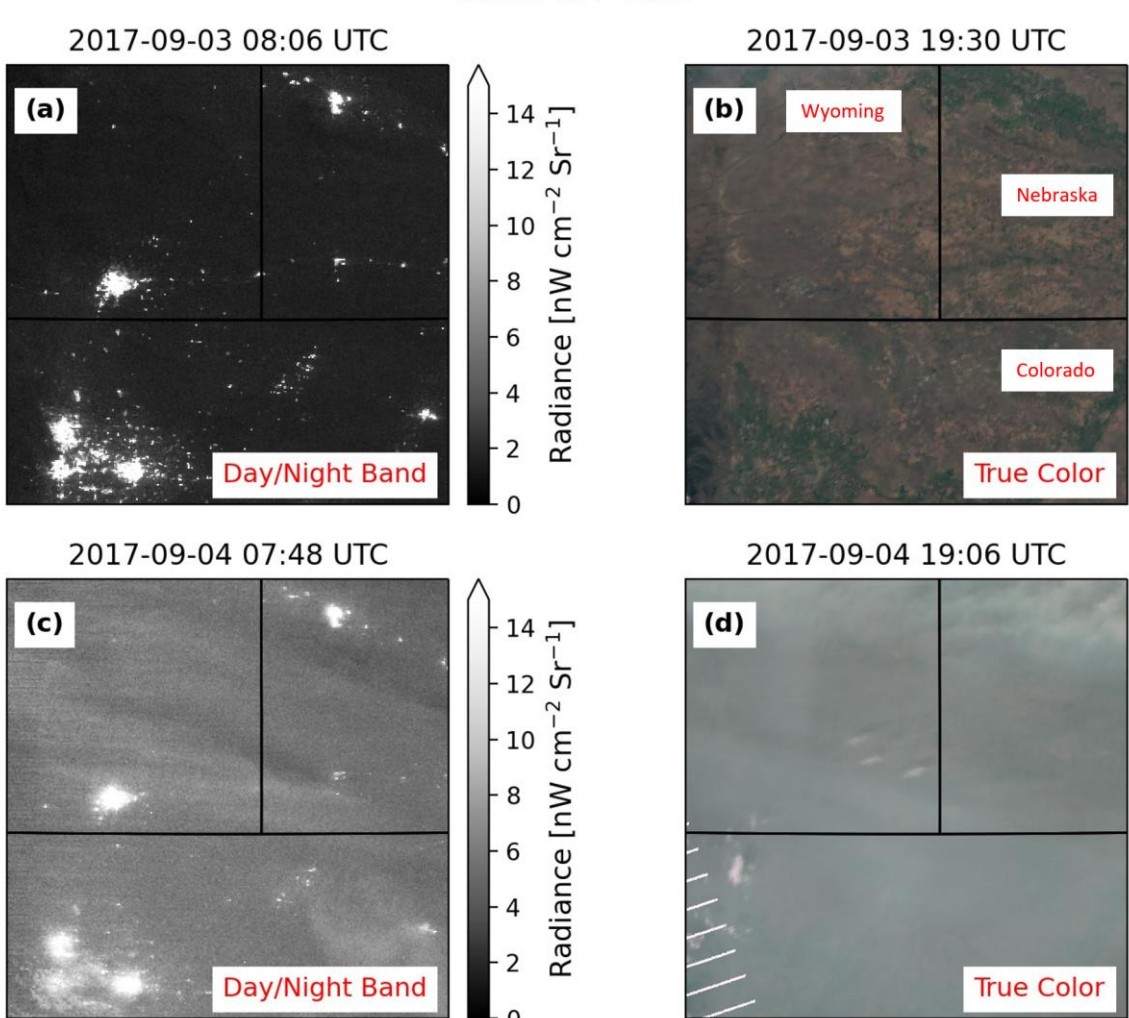

**Figure 1. a) VIIRS DNB image at 08:06 UTC on Sep. 03, 2017 over Central US. b) VIIRS true color image at 19:30 UTC on Sep. 03, 2017 for the same region as a). Both panels a) and b) correspond to relatively low aerosol loading in the scene. c) Similar to a)**
**but for the VIIRS DNB image at 07:48 UTC on Sep. 04, 2017. d) Similar to b) but for the VIIRS true color image at 19:06 UTC on Sep. 04, 2017. Both panels c) and d) correspond to relatively high aerosol loading (here, biomass smoke) in the scene, with commensurate blurring of city light structures when comparing c) to a). In both the clean and turbid DNB imagery examples, the Moon's phase is in waxing gibbous.**

Quantitatively, as suggested in Zhang et al. (2019), VIIRS/DNB received radiance (I) can be written as:

$$I = \frac{r(F_{direct}+F_{diffuse})+\pi I_a}{\pi(1-r_s\bar{r})}\left[e^{-\tau/\mu} + \mathrm{T}(\mu)\right] + I_p \,, \tag{1}$$

Here $r_s$ and $\bar{\bar{r}}$ are the reflectance for the surface and the aerosol layers, respectively, $F_{direct}$ and $F_{diffuse}$ are the direct and diffuse transmitted lunar fluxes to the surface, $I_a$ is the surface upwelling radiance from an artificial light source. μ is the





cosine of the lunar zenith angle, $\tau$ is the total optical thickness, including optical thickness of aerosol and gas molecules for

cloud-free skies, $e^{-\tau/\mu}$ and $T(\mu)$ are direct and diffuse transmittances in the line of sight direction, and $I_p$ is the path radiance.

The expression $\frac{r(F_{direct}+F_{diffuse})+\pi I_a}{(1-r_s\bar{r})}$ represents the surface upwelling energy from both reflected moonlight and light

emission from the artificial light sources, after accounting for multiply-scattered radiance reflected between the aerosol layer

and the surface layer (via a plane parallel radiative transfer assumption).

The $F_{direct}$, $F_{diffuse}$ and $I_p$ terms are difficult to obtain at the pixel level. Nevertheless, as suggested in Zhang et al. (2019), for a

given city, $F_{direct}$, $F_{diffuse}$ and $I_p$ terms remain relatively constant; however, the $I_a$ term has a large spatial variation. Thus, by

taking the spatial derivative of Equation 1, we readily obtain:

$$dI = \frac{dI_a}{1-\bar{r}r_s}\left[e^{-\tau/\mu} + T(\mu)\right],\tag{2}$$

Here $dI$ is the observed spatial gradient of artificial lights over a given city and can be directly estimated from the

VIIRS/DNB data; $dI_a$ is the spatial gradient of surface artificial light emissions of the city, which can be estimated using

VIIRS/DNB data over aerosol- and cloud-free conditions or through the use of VIIRS VNP46 data. Also, for a given

atmospheric state, the relationship between diffuse and direct transmittances can be calculated using a radiative transfer

model (RTM; e.g., Zhang et al., 2019).

We can derive optical thickness from Equation (2) as:

$$\tau = \mu\ln\frac{dI_a}{kdI(1-\bar{r}r_s)}\quad,\tag{3}$$

Here, $\overline{rr}_s$ accounts for the reflection of upwelling radiances back to surface and is assumed to be negligible in this study. The

$k$ term is a correcting factor to account for the difference between direct and diffuse transmittance, and is estimated using the

6S RT model (Vermote et al., 1997) for smoke, pollutant, and dust aerosols (see Zhang et al., 2019). A fine pollutant aerosol

was assumed for the USA region while dust aerosol is assumed for the Middle East and Indian Subcontinent regions. As for

the Indian Subcontinent, aerosol plumes primarily consist of a mixture of polluted haze, smoke and pollutant aerosols; we

perform a sensitivity study on this issue in Section 4.5. Here, $\tau$ is the total column optical thickness, which includes both

Rayleigh optical thickness and aerosol optical thickness for cloud-free skies. The AOT is then estimated by subtracting the

Rayleigh optical thickness (as calculated using 6S RT model calculations at 700 nm) from the total column optical thickness.

In Zhang et al. (2019), the spatial variation of VIIRS observed radiance (dI) and surface artificial light source emissions ($dI_a$)

are estimated empirically, using the standard deviation of VIIRS/DNB data over the observed aerosol-laden, cloud-free skies

($\Delta I$) and the standard deviation of VIIRS/DNB data over aerosol- and cloud-free skies ($\Delta I_a$), for a given artificial light source

(hereafter, we refer to this process as the empirically-based STD method). Besides using the standard deviation-based

method for estimating spatial variability, we also explore here the feasibility of applying two other methods for estimating

the spatial gradient. The first method is referred to as the "Mean Method". For this method, the VIIRS/DNB data over a

given artificial light source are sorted, and the difference between means of the brightest 50% of the data and the darkest



50% of the data within a defined grid is used to represent dI. The same method is applied to VIIRS/DNB data over cloud and aerosol free skies to estimate $dI_a$ for a given light source. The second proposed method is referred to as the "Median Method". This method is similar to the "Mean Method" but uses median values instead of mean values for the 50% brightest and 50% darkest VIIRS DNB data over the given artificial light source.

**3.2 Construction of an equal-grid space for AOT retrievals**

In contrast to the approach adopted by Zhang et al., (2019), wherein aerosol retrievals were performed for selected cities, the spatial domains in this evaluation are divided into 25 x 25 km² equal-area grids, and retrievals are performed on the entire domain of these grids. The advantage of this approach is that *a priori* knowledge of city/town locations is no longer needed. Also, by performing retrievals at the grid level, the sampling bias related to city/town sizes and densities is also reduced. Retrievals in this study were conducted for three regions (Figure 2): the USA, the Indian Subcontinent, and the Middle East.

For the USA, the region center was set at 37°N 97°W, with selected domain width and height of 4700 km and 2700 km, respectively. For the Indian Subcontinent region, the center was set at 20°N 78°E, with a domain width of 3420 km and grid height 3500 km. For the Middle East region, the center was set at 30°N 45°E, with a domain width of 4200 km and a height of 4200 km. No off-shore retrievals were considered for this study, although there exist numerous anthropogenic light sources (which tend to occur as points, by nature of them typically being either boat lights of offshore drilling platforms).

The ephemeral nature of migratory boat lights presents an inherent challenge, but offshore drilling platforms may provide a steady and well-defined signal for nighttime AOT applications; a good candidate for future research.

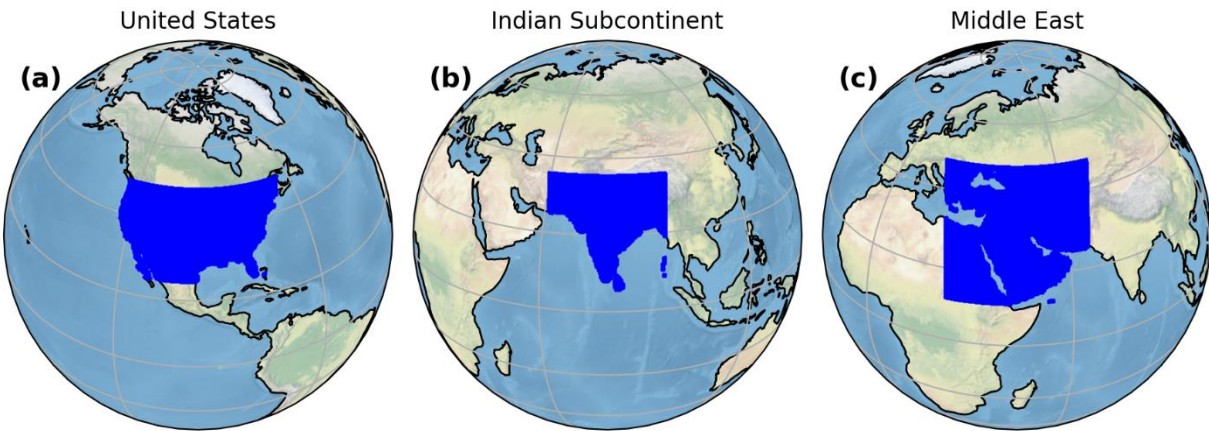

**Figure 2. a) Highlighted in a bright blue color are the selected grid regions (each grid is 25x25 km² equal) for the USA region. b) Similar to a) but for the Indian Subcontinent (SC) region. c) Similar to a) but for the Middle East region.**

**3.3 Cloud screening and quality assurance steps**

As in Zhang et al. (2019), potential artificial light sources are initially selected by identifying cloud-free, quality-controlled VIIRS pixels that have radiance values 1.5 times larger than surrounding cloud-free background radiance values. The VIIRS



VCCLO data are used for cloud clearing of VIIRS data. Since the spatial resolution of VIIRS DNB radiance data is double that of the VIIRS cloud mask data, the latter were resampled (i.e., oversampled) to match the spatial resolution of the VIIRS
DNB radiance data during the cloud-clearing process. Solar-contaminated pixels as well as low quality pixels, as indicated by the VIIRS DNB QA flags (such as pixels with bad calibration, or pixels with unreliable data readings), were also excluded following the procedure listed in Zhang et al., (2019).

To further exclude cloud-contaminated pixels, two additional tests, described in detail by Zhang et al., (2019) were implemented. In the first test, we assume that averaged mean geolocations for artificial lights within a given 25 x 25 km$^2$ grid
remain unchanged over cloud-free nights. Thus, yearly mean latitude and longitude for artificial lights within a given 25 x 25 km$^2$ grid were computed, and nights with mean geolocations that were more than 0.02° (latitude/longitude) away from the yearly means were excluded. Here we assumed that on some cloudy nights, when artificial lights may be strongly attenuated and become undetectable, the mean geolocations for detected artificial lights on a given night within a given grid box may differ from yearly mean-computed values possibly due to spatial variations in daily cloud coverage. Also, VIIRS
pixel counts for nights that passed the above-mentioned geolocation check were recorded, and the night with the minimum VIIRS pixel count was earmarked. In later processes, for a given light source, only those nights where the number of detected artificial light source pixels was greater than this minimum value were considered for further analysis.

For the second test, the correlated relationship between daily mean radiance and standard deviation of radiances was used for additional cloud-clearing and for removal of bad data samples (Zhang et al., 2019) through a two-step screening approach.
For the first screening step, yearly mean and standard deviation of radiances from artificial lights for a given grid were computed based on the daily mean and standard deviation of radiances of the grid. For a given night, if the daily standard deviation of radiances was larger than the yearly mean plus twice the yearly standard deviation, then that night was assumed to be either cloud-contaminated or to consist of bad data. This step, however, could exclude heavy aerosol loading cases by misclassifying heavy aerosol plumes as clouds. A linear relationship between daily mean radiance and standard deviation of
radiance was constructed for each grid using data points that passed the first screening check. This linear relationship was used to predict daily standard deviation of radiances from the daily mean radiance value from pixels with artificial lights. The updated yearly mean of standard deviations was also computed. For a given grid cell and for a given night, if the actual daily standard deviation of radiances was larger than the predicted standard deviation of radiances for that night plus half the updated yearly mean standard deviation, then those data points were considered as either cloud-contaminated or bad values.
Note that the above-mentioned QA approaches were initially designed for the STD method. Here, we have adopted similar approaches for the Mean and Median methods.

### 3.4 Identifying surface artificial light source emissions using empirical methods

As suggested in Equation 3, the spatial derivative of artificial light source emissions for a given grid cell is needed for nighttime AOT retrievals. The spatial derivatives of artificial light source emissions are derived using four methods in this
study – three that are empirically based (STD, Mean, and Median) and the other based on NASA VNP46 data. The first





empirical approach was adopted from Zhang et al., (2019), in which the standard deviation of surface artificial light emissions for a given region, or $\Delta I_a$, was used to represent the spatial derivative of an artificial light source. This method is referred as the empirical-based STD method. The other two empirical approaches estimated the difference in mean or median values of the brightest and darkest 50% artificial light sources (light emissions sources only), or $dI_a$ values, respectively. For the VNP46-based approach, $\Delta I_a$, values, or the standard deviations of artificial light source emissions, were estimated using the NASA VNP46 data (VNP46-based STD method).

For the empirical-based STD method, we assumed the standard deviation of radiances from artificial light sources for a given grid was higher over an aerosol- and cloud-free night than over nights with aerosol or cloud contamination. Thus, for each grid, we picked 30% of nights that had the largest standard deviation values, and computed the mean ($\Delta I_{a\_mean}$) and standard deviation ($\Delta I_a\_std$) of the standard deviation of radiance values for those nights.

The averaged MISR AOT (550 nm) for 2017 for the DJF, MAM, JJA, and SON seasons were (respectively) 0.06, 0.11, 0.14, and 0.09 for the USA, 0.26, 0.36, 0.28, and 0.26 for the Indian Subcontinent, and 0.14, 0.27, 0.33, and 0.22 for the Middle East region. The averaged MODIS AOT (550 nm) for the DJF, MAM, JJA, and SON seasons for 2017 were (respectively) 0.07, 0.12, 0.17 and 0.14 for the US, 0.32, 0.40, 0.44 and 0.33 for the Indian Subcontinent, and 0.10, 0.18, 0.24 and 0.16 for the Middle East region. Thus, on average, we considered the USA as a relatively clean particulate matter-region, the Middle East as a moderately polluted particulate matter region, and the Indian Subcontinent as a heavily polluted region. Further, we assumed $\Delta I_a$ values to be $0.9*\Delta I_{a\_mean}$, $\Delta I_{a\_mean}$, and $1.1*\Delta I_{a\_mean}$ for clean, moderately aerosol polluted, and heavily aerosol polluted regions, respectively, assuming $\Delta I_a$ values are underestimated over heavy aerosol polluted regions. The sensitivity of AOT retrievals to $\Delta I_a$ values is discussed in Section 4.3.

A new QA step was also implemented to monitor changes in artificial light source patterns for further removal of cloud-contaminated and bad data. In this new step, for a given night and for a given grid, the mean distance of all detected artificial light sources to the most south-western data point was computed. The yearly mean and standard deviation of the daily mean distance were also computed. For smaller cities, or grids with fewer than 100 identified artificial light source VIIRS pixels, if the variation of the daily mean distance (yearly standard deviation in mean distance divided by yearly averaged mean distance) was larger an 25%, then the artificial pattern within a given grid is considered unstable and thus is then excluded from the analysis. This approach can be viewed as a simplified method for checking changes in artificial light patterns from night to night.

For the empirical-based Mean and Median methods, similar approaches were implemented. Here, for a given grid, we simply replaced the standard deviation of radiances ($\Delta I_a$) from the artificial light pixels either with the difference ($dI_a$) in *mean* radiances of the brightest 50% and the darkest 50% artificial light pixels, or with the difference in *median* values of the brightest 50% and the darkest 50% artificial light pixels. For each grid, we picked 30% of nights that had the largest $dI_a$ values, and computed the mean ($dI_{a\_mean}$). We again assumed $dI_a$ values to be $0.9*dI_{a\_mean}$, $dI_{a\_mean}$, and $1.1*dI_{a\_mean}$ for clean, moderately polluted, and heavily polluted regions (i.e. USA, Middle East and Indian Subcontinent), respectively.



## 3.5 Identifying surface artificial light source emissions using NASA's Black Marble data

As an alternative to the empirical approach, the NASA VNP46 data provides estimation of atmospheric and surface BRDF effect corrected surface light source emissions (Román et al., 2018). Zhang et al., (2023) suggest that the NASA VNP46 Black Marble data may be used for estimating $\Delta I_a$ (artificial light spatial change) values. In this study, we examined the feasibility of using level 3 monthly NASA VNP46 data as $\Delta I_a$ for the STD-based method. We estimated yearly $\Delta I_a$ values using monthly VNP46 data.

In this approach, monthly VNP46 data within a 50 x 50 km$^2$ area over a given 25 x 25 km$^2$ grid were obtained at the raw VNP46 data resolution. At each grid point on each night, monthly VNP46 data were collocated with VIIRS DNB data at the pixel level based on the nearest-neighbor method (e.g. picking the closest VNP46 data in latitude/longitude for a given VIIRS/DNB pixel). For a given pixel on a given night, similar approaches were conducted using monthly VNP46 data from all 12 months, and then averaged to construct the yearly mean VNP46 data for the given pixel location. The daily $\Delta I_a$ value

was then derived from the yearly mean of monthly VNP46 data for all identified night-light sources (which may vary on a daily basis) for a given 25 x 25 km$^2$ grid for each night. Note that the VNP46-based mean and median methods were not implemented for reasons mentioned below in Section 3.5.

## 3.6 Inter-comparison of the empirically-based and VNP46 data based estimation of ΔIa

Figure 3 shows the comparison between $\Delta I_a$ values that were derived using the empirically-based STD method as described

in Section 3.3 and the NASA VNP46 Black Marble data as described in Section 3.4. As reported by Zhang et al. (2019), the TOA VIIRS DNB radiance is a strong function of viewing angle. As a result, the viewing angle correction, as described in detail in Zhang et al. (2019) has been implemented in both approaches.

The grey symbols in Figure 3 represent collocated pairs of $\Delta I_a$ values from the two methods for each grid cell and for each night. Note that for a given grid cell, only one $\Delta I_a$ value is estimated for a given year. $\Delta I_a$ values from the NASA VNP46

data change on a daily basis, as detected city light pixels vary on a daily basis for a given grid cell. Thus one $\Delta I_a$ value from the empirical method may be associated with up to hundreds of $\Delta I_a$ values from the VNP46 data. Also, for the empirical-based STD method, $\Delta I_{a\_mean}$, was used in the comparison for all three regions, with no correction applied for aerosol loading, for the $\Delta I_a$ values from the empirical method, as mentioned in Section 3.3.

The correlations of $\Delta I_a$ values from the two methods were 0.9, 0.91 and 0.89 for the USA, Middle East, and Indian

Subcontinent regions, respectively, with associated regression slopes at 0.63, 0.48 and 0.42, and the RMSE values are 0.85, 2.18 and 1.22 (x10$^{-8}$ W cm$^{-2}$ sr$^{-1}$) . We also averaged daily $\Delta I_a$ values from the VNP46 data on a yearly basis. The pairs of yearly $\Delta I_a$ values from the two methods are shown with red symbols in Figure 3.



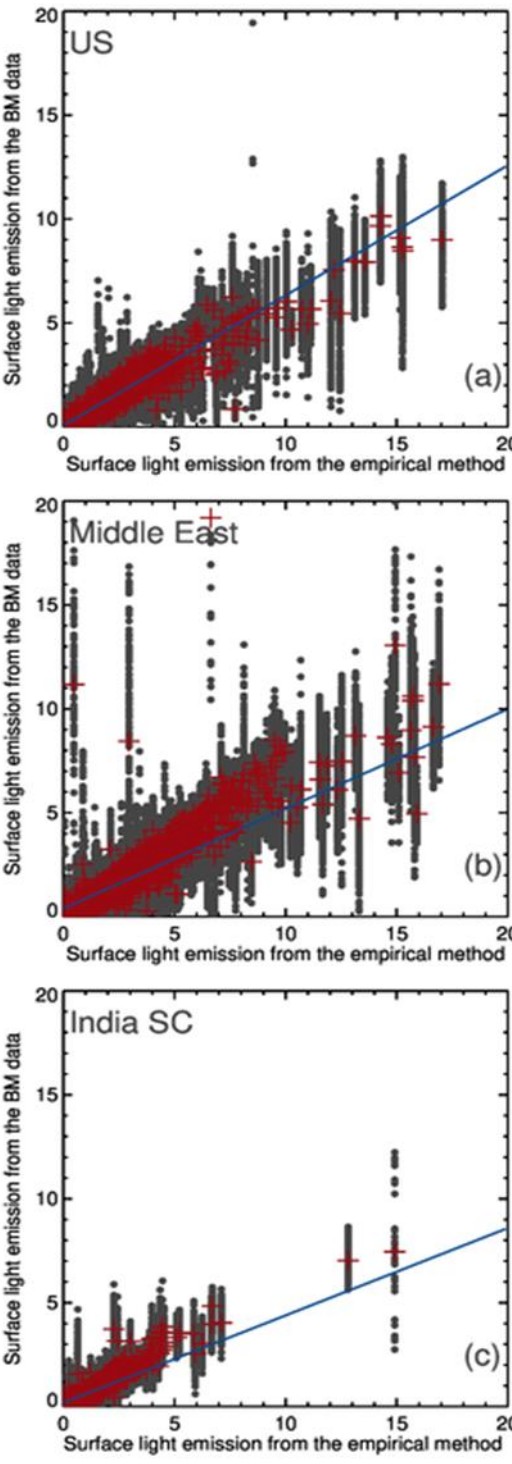

**Figure 3. Scatter plots of $\Delta I_a$ values (in unit $10^{-8}$ W cm$^{-2}$ sr$^{-1}$) estimated from the empirically-based method and the NASA's VNP46 data (grey entries). The blue lines are the linear fits through the paired data sets. Red crosses show the similar analysis but with**



the use of yearly averaged $\Delta I_a$ values from the NASA's VNP46 data. a) USA domain; b). Middle East domain; c) Indian Subcontinent domain.

The high correlation values of around 0.9 of $\Delta I_a$ values between the two methods for all three regions show that $\Delta I_a$ values from both methods are strongly related. However, some important differences between the regions were noted. The largest

slope was found over the USA region and the smallest slope was found over the Indian Subcontinent. As noted in Section 3.3, the USA is only lightly polluted in non-summer months, while the Indian Subcontinent region is heavily polluted throughout the year, but with biomass burning peaking in late fall, significant polluted haze in winter, winter and additional dust in spring. Thus, it is suspected that in the presence of such heavy aerosol pollution, the NASA VNP46 data may be low-biased in estimating aerosol-free $\Delta I_a$ values. It is also possible that the low bias is caused by the use of yearly mean of

monthly VNP46 data, which is the approach suggested in Zhang et al. (2023). For the Indian Subcontinent or the Middle East regions, even the $\Delta I_a$ values estimated from the empirical methods may be underestimated as well.

Per Equation 2, any low biases in $\Delta I_a$ will invariably introduce low biases in retrieved AOTs and hence no attempt was made to retrieve nighttime AOTs using $\Delta I_a$ values derived from the VNP46-based STD method for the Middle East and the Indian Subcontinent regions. We expected a similar low bias for the "Mean" and "Median" methods using the NASA's VNP46

data for these two regions as well. However, over relatively aerosol-free regions (based on yearly averages, not individual events) such as the USA region, it was anticipated that NASA's VNP46 data would indeed by useful for application to nighttime AOT retrievals.

## 4 Results and Discussions

In this section, retrieved AOTs from the proposed satellite-based methods are inter-compared with surface-based

daytime/nighttime AERONET data. The spatial distributions of VIIRS/DNB nighttime AOTs are also inter-compared with daytime MISR AOT distributions at the seasonal average scale.

### 4.1 Evaluation of nighttime AOT retrievals from the empirically- and VNP46-based STD method using the nighttime AERONET AOT data

Using lunar AERONET data, we evaluated the performance of nighttime AOT retrievals from the empirically-based and

VNP46-based STD methods for the USA region. Due to differences in data screening during the QA process, there were 892 collocated pairs available for the empirically-based method (Figure 4a) and 837 collocated pairs for the VNP46-based method. Figure 4a shows the comparison between lunar AERONET AOT (675 nm) and VIIRS/DNB retrieved AOT for the USA region for 2017. Correlation and RMSE values of 0.81 and 0.13 are indicated in Figure 4a. Comparable values of 0.76 and 0.14 are shown in Figure 4b, in which nighttime AOT retrievals were performed using the VNP46-based STD method.

This favorable comparison was expected, as the USA region is relatively aerosol-free for most days in the year; thus, Figure



4b suggests that yearly averaged monthly NASA VNP46 data can be used to represent $\Delta I_a$ values for nighttime aerosol retrievals for the USA region, with an estimated noise floor of ~0.15.

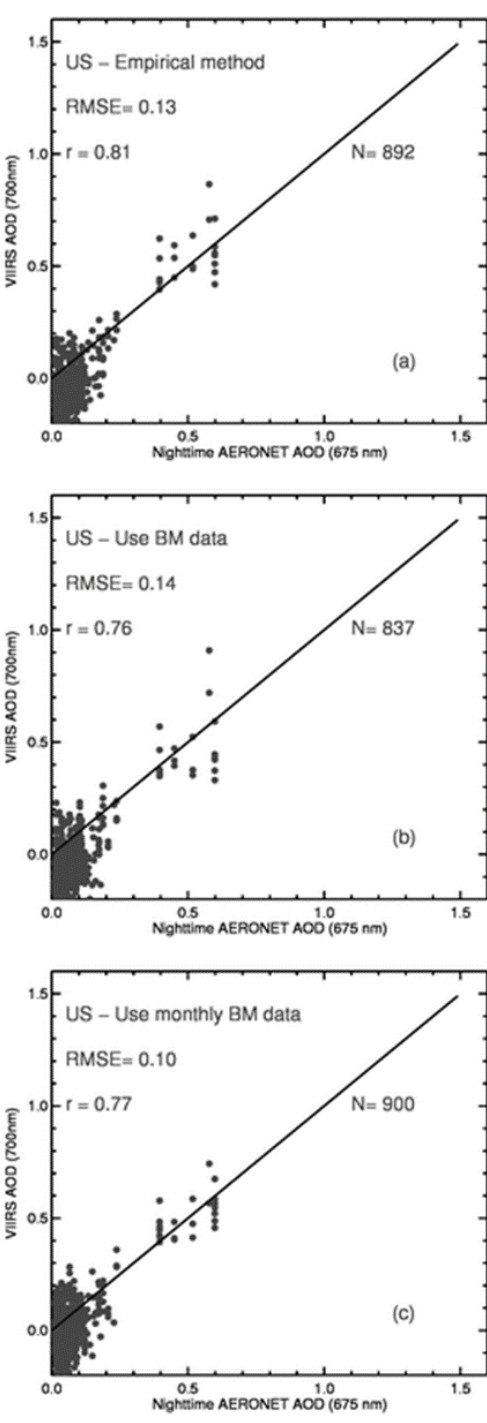





**Figure 4. a) Scatter plot of nighttime AOTs derived from VIIRS/DNB (700 nm) and AERONET (675 nm) over the USA region.**
**$\Delta I_\mathrm{a}$ values were estimated using the empirically-based method. Also shown is the one-to-one line. b) Similar to a) but using the yearly averaged monthly NASA's VNP46 data to estimate $\Delta I_\mathrm{a}$ values. c) Similar to b) but using the monthly NASA's VNP46 data to estimate $\Delta I_\mathrm{a}$ values.**

The $\Delta I_\mathrm{a}$ values used in AOT retrievals in Figure 4b were derived using the *yearly* mean of the monthly VNP46 data. We also attempted this approach using *monthly* NASA VNP46 data for representing $\Delta I_\mathrm{a}$ values for a given month, and these results are shown in Figure 4c. A reduced RMSE value of 0.10 was found, with a correlation value of 0.77 observed, suggesting that the use of monthly VNP46 data may offer a slightly better result.

Figure 4 seems to suggest that for regions that are mostly low aerosol loading on a yearly basis both the empirical-based and VNP46-based STD methods with NASA's VNP46 data can be used for nighttime AOT retrievals with reasonable accuracy. Although not shown, we observed low biases in satellite retrieved nighttime AOTs over the Middle-East (with a linear offset of -0.13, and a RMSE of 0.27) and the Indian Subcontinent (with a linear offset of -0.44 and a RMSE of 0.56) aerosol polluted regions. This finding was expected, as we suspect that aerosol contamination may exist in NASA's VNP46 data over regions with frequent aerosol pollution events. We address this topic in results to follow.

## 4.2 Evaluation of the nighttime VIIRS/DNB AOT retrievals from the empirically-based STD, mean and median methods using the daytime and nighttime AERONET AOT data

In this section, we compare nighttime VIIRS/DNB AOTs from the three empirically-based methods (STD, Mean and Median) against both nighttime and daytime AERONET data. Figure 5a shows the comparison between nighttime VIIRS AOT (700 nm) values derived from the empirically-based STD Method for the combined USA, Middle East and Indian Subcontinent regions, versus lunar AERONET AOT values at 675 nm. A total of 1119 collocated AERONET / VIIRS pairs were available for this analysis, with a correlation of 0.82, RMSE 0.14, slope of 1.13 and an offset of -0.11. Figure 5d is similar to Figure 5a, but with comparisons made against daytime AERONET data. A total of 24861 collocated pairs were found, with a correlation of 0.61, RMSE 0.18, slope 0.80 and an offset of 0.06 between VIIRS/DNB (nighttime) and AERONET (daytime) AOT.

In contrast, with a totality of 1122 collocated pairs, a slope value of 1.14, RMSE 0.14, correlation 0.84 and an offset of -0.12 were found between nighttime AERONET AOT and nighttime VIIRS/DNB AOT derived using the empirically-based Mean Method. The statistics are similar to those derived using the empirically-based STD method, suggesting both methods can be used for nighttime AOT retrievals. This suggestion is further corroborated by the comparison of daytime AERONET AOT and nighttime VIIRS/DNB AOT derived using the empirically-based Mean Method as shown in Figure 5e. In comparison with Figure 5d, for which AOTs are derived using the STD Method, slight improvements in the correlation from 0.61 to 0.68, slope from 0.80 to 0.95 and offset from -0.12 to -0.09 were found, along with a reduction of RMSE of 10% and a data loss of ~13%. This data loss is again a result of QA screening, as mentioned in Section 3. It appears that the empirically-based Mean Method performs better, although some heavy aerosol loading cases (e.g. AERONET AOT > 0.7 at 675 nm), as shown in Figure 4d, were excluded during QA, possible related to misclassification in cloud masking.

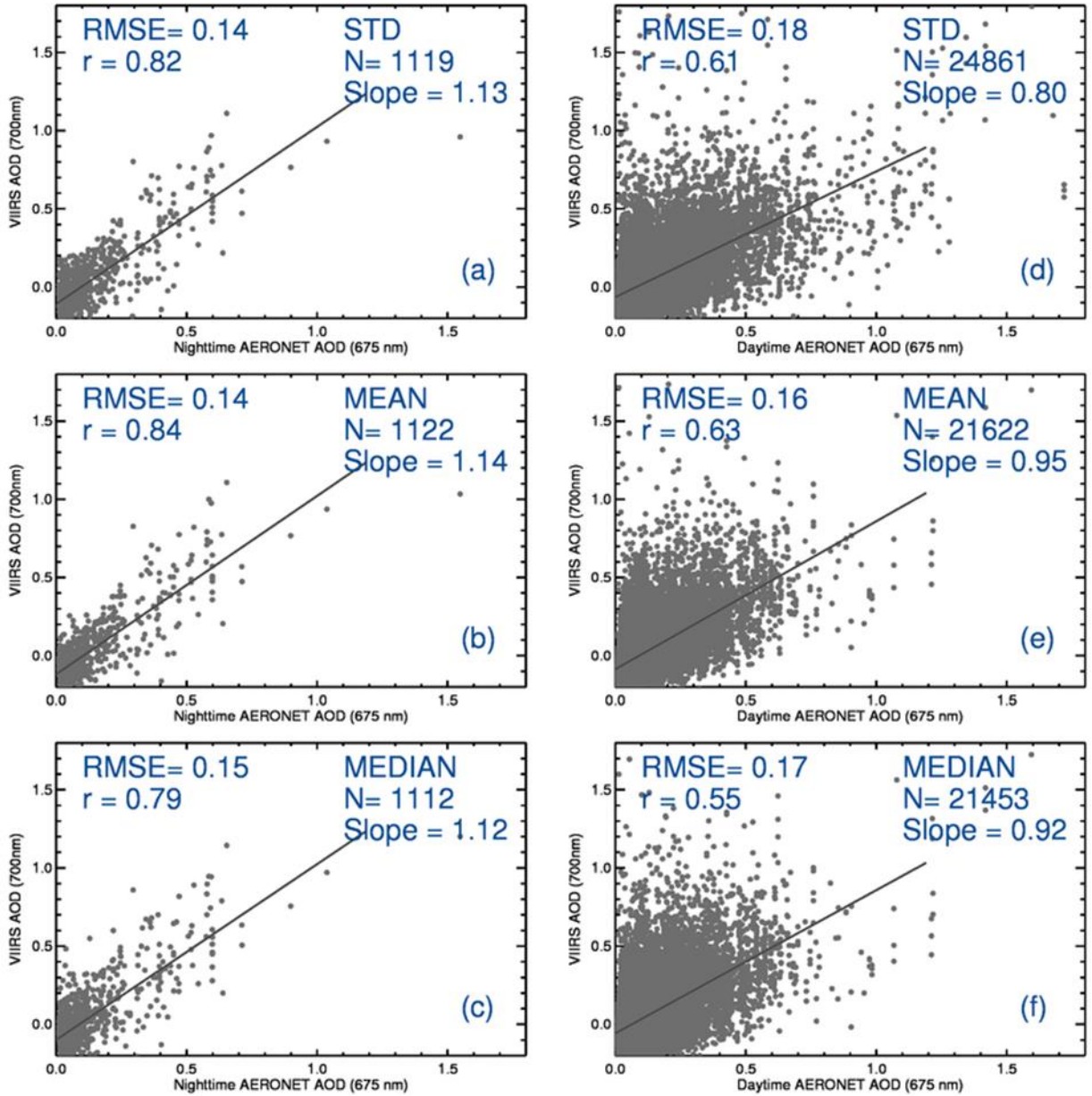

**Figure 5. a) Scatter plot of nighttime AOTs derived from VIIRS (700 nm) by the empirically-based STD method and nighttime**
**AERONET (675 nm) over the USA, the Indian Subcontinent and the Middle East regions. d) Similar to a) but with the use of**
**daytime AERONET data. b) and e) Similar to a) and d) but with use of the empirically-based Mean method. c) and f) Similar to a)**
**and d) but with empirically-based Median method.**

The performance of the Median Method was slightly less optimal than the empirically-based Mean Method. For this

comparison with nighttime AERONET AOTs, a correlation of 0.79, slope 1.12, RMSE 0.15 and an offset of -0.10 were





found.  For the comparison with daytime AERONET AOTs, the correlation, slope, offset  and RMSE values were 0.55, 0.92, -0.06 and 0.17 respectively.

Figure 5 suggests that both the dI (from the Mean and Median empirically-based Methods) and ΔI (from the STD Method) values contain some aerosol-related information content and can be used for nighttime AOT retrievals.

### 4.3 Parameter quantification for nighttime aerosol optical thickness retrievals

**4.3.1 The impact of grid cell size**

In constructing Figures 4 and 5, we required the minimum number of artificial light source VIIRS pixels for a given grid cell to be larger than 50, as suggested from a previous study (McHardy et al., 2015), and the mean number of VIIRS pixels for a given city to be larger than 60. However, it is potentially insightful to investigate the impact of city size on nighttime VIIRS AOT retrievals. In this evaluation (Figure 6a), the pixel requirement was removed and cities with a minimum number of

VIIRS pixels less than 50 were included. The differences between the resulting nighttime VIIRS/DNB (700 nm) and AERONET (675 nm) AOTs (AOT$_{diff}$) were plotted against the number of VIIRS pixels for USA cities, using the empirically-based STD Method.  Unsurprisingly, AOT$_{diff}$ values had a much larger data spread of 0.8 for cities with size around 50 VIIRS pixels or less, and a spread of around 0.2 for cities with size above 500 VIIRS pixels.  As suggested from McHardy et al. (2015), with the increase in both the city size and the number of pixels for a given city, the derived $\Delta I_a$ and

$\Delta I_{sat}$ values are more stable with larger SNR values, and are thus more suitable for nighttime aerosol retrievals. Similar results were found for retrieval of VIIRS AOT using the empirically-based Mean Method (Figure 6b), indicating the importance of keeping the pixel number requirement in the retrieval.

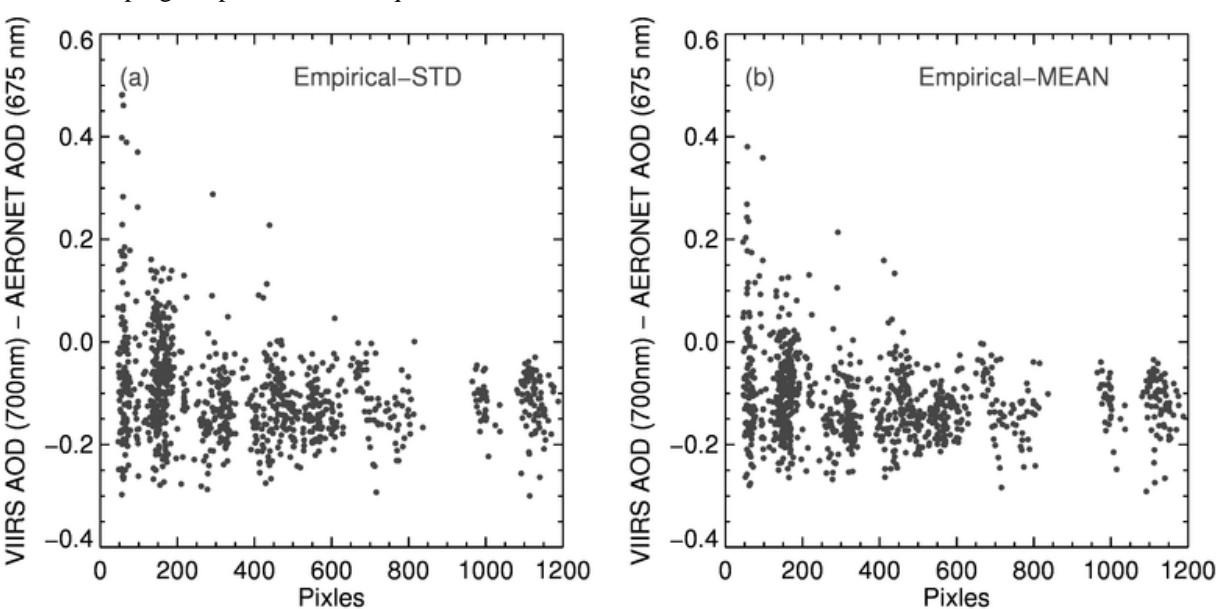





**Figure 6. The difference between VIIRS/DNB (700 nm) and lunar AERONET (675 nm) AOT as a function of the number of**
**detected artificial light source pixels in a grid, using the empirically-based STD method (a) and the empirically-based Mean**
**method (b).**

### 4.3.2 The impact of the spatial derivative of surface light source emissions

As indicated from Equation 3, in order to derive nighttime VIIRS/DNB AOT values, accurate estimates of the spatial gradient of surface artificial emissions ($\Delta I_a$ for the STD Method and $dI_a$ for the Mean and Median Methods) were needed.

For each of the empirically-based Methods, the spatial gradients of surface artificial emissions were estimated using relatively clear aerosol- and cloud-free nights.

For the empirically-based STD Method, we tested the sensitivity of AOT retrievals to $\Delta I_a$ values by using 0.9 x $\Delta I_a$, 1.0 x $\Delta I_a$ and 1.1 x $\Delta I_a$ as the standard deviations for artificial light source emissions over aerosol and cloud-free skies. As shown in Figure 7a, where adjusted $\Delta I_a$ values are applied for AOT retrievals for the three selected regions, higher values of retrieved

AOT are associated with higher $\Delta I_a$ values. This finding is not surprising, as it can be theoretically explained using Equation 3. Nevertheless, it is worth noting that with the use of higher $\Delta I_a$ values, high biases in VIIRS/DNB AOT are found at the low AERONET AOT ranges as shown in Figure 7a (red dots). Similar results are found by repeating the same exercise using the VIIRS/DNB AOT data from the empirically-based Mean Method (Figure 7b).

For the Indian Subcontinent region, a desert aerosol model was used in place of urban aerosol models when accounting for

diffuse transmittance. In reality, the Indian Subcontinent hosts a wide variety of aerosol species. This situation provides for a natural laboratory for the sensitivity of the methods for the optical model chosen. Figure 7c shows differences between nighttime VIIRS/DNB AOT (700 nm) and lunar AERONET AOT (675 nm) comparing the desert and urban aerosol models for the empirically-based STD Method. Similar results are shown in Figure 7d but for using the empirically-based Mean Method. In both cases, the difference in VIIRS AOT is near zero for low aerosol-loading cases (as indicated by lunar

AERONET data) and approaches 0.1 for the lunar AERONET AOT values near 1. This excise suggests that the proposed nighttime AOT retrievals may be less sensitive to aerosol model used for the correcting factor k as indicated in Equation 3.



**Figure 7. a)** Scatter plot of lunar AERONET and VIIRS/DNB AOT as a function of $\Delta I_a$ value for the USA region using the empirically-based STD method. **b)** Scatter plot of lunar AERONET and VIIRS/DNB AOT as a function of d$I_a$ value for the USA region using the empirically-based Mean method. **c)** Scatter plot of lunar AERONET and VIIRS/DNB AOT (derived using the empirically-based STD method) over the Indian Subcontinent region using different aerosol models. **d)** Similar to c) but with the use of the empirically-based Mean method.





### 4.3.3 Sensitivity of nighttime AOT retrievals as a function of observing conditions

Using differences in collocated daytime AERONET and nighttime VIIRS/DNB AOT data, we also examined the sensitivity of the nighttime VIIRS/DNB AOT retrievals to observing conditions such as lunar fraction, sensor zenith angle and Julian day. We use daytime AERONET data, since the number of available collocated daytime AERONET and VIIRS data pairs is significantly higher than the number of available collocated nighttime AERONET and VIIRS data pairs, as suggested in Figure 5. Also please note that we are essentially compare nighttime AOT retrievals with daytime AOT retrievals, and there

are non-negligible diurnal variations in AOTs, especially for significant aerosol events. Also, while comparing nighttime VIIRS with nighttime AERONET data, the results are biased towards cloud free skies as AERONET data are cloud screened, this bias doesn't exist for comparing nighttime VIIRS with daytime AERONET data. For all three regions, no apparent trend is found between the difference in daytime AERONET (675 nm) and nighttime VIIRS (700 nm) AOT and lunar fraction (Figures 8a-c), pointing in part to the stability of the lunar irradiance models employed (e.g., Miller and

Turner, 2009). Further, no major VIIRS AOT retrieval biases were found as a function of sensor viewing angle or Julian day for the USA and Middle East region. Figures 8a-8c and 8g and 8h appear to indicate that nighttime AOT can be effectively retrieved regardless of moon conditions (which are themselves a function of Julian day).

A sensor zenith angle trend was apparent in the Indian Subcontinent region, and retrievals from summer months were significantly reduced in number, likely due in part to summer monsoonal cloudiness. We suspect that this result occurs

because the Indian Subcontinent region is heavily polluted all year round, and thus the sensor zenith angle correction (Zhang et al., 2019), may be less applicable for this region. Regarding the smaller number of summer retrievals in the India region, we found that detected artificial light source sizes reduce to minimum levels during the summer season especially for smaller size cities/towns with less than 200 VIIRS pixels; a similar phenomenon was not observed with the other two regions. This shrinking in detected artificial light source sizes causes some summer data to be removed from the retrieval during the QA

process. The Indian Subcontinent region is generally polluted by aerosols, and is frequently cloud covered; thus, fewer artificial light pixels are detected. It is also possible that with the stray-light correction (Mills and Miller, 2016) that moves down to mid-latitudes during the summer, correction and its associated errors in subtracting-out background noise is disproportionately compared to the other regions. We leave this topic for another study.



**Figure 8. a) The difference between daytime AERONET (675 nm) and nighttime VIIRS/DNB (700 nm) AOT as a function of lunar fraction (%) for the USA region using the empirically-based STD (black color) and mean (blue) colors. d) Similar to a) but showing the difference between daytime AERONET (675 nm) and nighttime VIIRS/DNB (700 nm) AOT as a function of sensor viewing zenith angle. g) Similar to a) but showing the difference between daytime AERONET (675 nm) and nighttime VIIRS/DNB (700 nm) AOT as a function of Julian day. b, e, and h) Similar to a, d and g) but for the Middle East region. c, e and i) Similar to a, d and g), but for the Indian Subcontinent region.**

## 4.4 Regional retrievals

It is also interesting to inter-compare nighttime VIIRS/DNB AOT retrievals with other spatial collections of aerosol observations, for example from satellite AOT retrievals made during the daytime. Figures 9a - 9d show seasonal averaged daytime AOT retrievals (550 nm) from MISR for the DJF, MAM, JJA, and SON seasons, respectively. Level 2 (version 23) MISR AOT data were averaged into 0.5° x 0.5° latitude/longitude grids for this exercise. Figures 9e - 9h show seasonally-averaged daytime dark target (DT) AOT retrievals (550 nm) from Aqua MODIS for the same four seasons, with Figures 9i -





9l showing the Aqua MODIS deep blue (DB) AOT retrievals (550 nm) for the same seasons. Although apparent differences can be found between MODIS DT, DB and MISR retrievals (due to differences in retrieval schemes and data sampling), AOT values are rather low over the USA for 2017 for all seasons, with locally higher AOT values found only for the SON

season.

Figure 9. a - d). **Seasonally averaged daytime MISR AOT (550nm) over the USA region for the DJF, MAM, JJA, and SON seasons. e – h) Similar to a – d), but for MODIS Dark Target (DT) AOD (550 nm). i – l) Similar to e – h), but for MODIS Deep**



**Blue (DB) AOD (550 nm). m - p) Similar to i - l) but for nighttime VIIRS/DNB AOT derived using the empirically-based STD method for estimating $\Delta I_a$ values. q - t) Similar to m - p) but using the empirically-based method for estimating $\Delta I_a$ values. u – x) Similar to q – t), but using nighttime VIIRS/DNB nighttime AOT derived through use of the NASA VNP46 data for estimating $\Delta I_a$ values.**

Figures 9m - 9p present the 1° x 1° (latitude/longitude) averages of nighttime VIIRS/DNB AOT (700 nm) for the same seasons as in Figures 9a - 9d, using nighttime VIIRS/DNB AOT retrievals from the empirically-based STD Method. Only

25 x 25 km$^2$ grids possessing at least 50 identified VIIRS/DNB artificial light pixels and having a mean value of more than 60 identified VIIRS/DNB artificial light pixels were included in the averages. Similar to Figures 9m - 9p are Figures 9q - 9t, which show the nighttime VIIRS/DNB AOT retrievals using the empirically-based Mean Method. Figures 9u - 9x show similar plots as Figures 9m - 9p, but with the use of nighttime VIIRS/DNB AOT retrievals from the VNP46 empirically-based STD Method. VIIRS nighttime retrievals from all three empirically-based Methods suggest that at the seasonal

average level, nighttime VIIRS/DNB AOTs are rather low for all seasons.

Despite these generally low nighttime VIIRS/DNB AOT levels, heavy aerosol events can be observed on the daily scale for these retrievals. Figures 10a - 10d show Aqua MODIS true color images for 3-6 September 2017, obtained from the NASA Worldview website (https://worldview.earthdata.nasa.gov/), with smoke plumes clearly visible over much of the USA. On 3 Sep 2017, smoke plumes were mostly located in the central US, with some plumes observable over the western US. On 4

Sep 2017, concentrated smoke plumes from the northwestern USA were transported to the north central US. On 5 Sep 2017, similar smoke patterns were observed, with smoke plumes advected into cloudy regions associated with a mid-latitude cyclone. On 6 Sep 2017, plumes were largely concentrated over the western US, while the middle and eastern USA were largely smoke-free.

The nighttime VIIRS/DNB AOT patterns show similar spatial patterns to their corresponding daytime MODIS true color and

the nighttime VIIRS DNB radiance imagery (Figure 10e – 10h), for both STD-based retrievals using $\Delta I_a$ values from either the empirical approach (shown in Figure 10i – 10l) or the NASA VNP46 data (shown in Figure 10m – 10p). It is interesting to note that the moon fraction changes from ~0.9 on 3 Sep to near 1 (full moon) on 6 Sep (e.g. Figure 10e - 10h), while AOT retrieval density as shown in Figures 10i-p is generally independent of moon conditions.

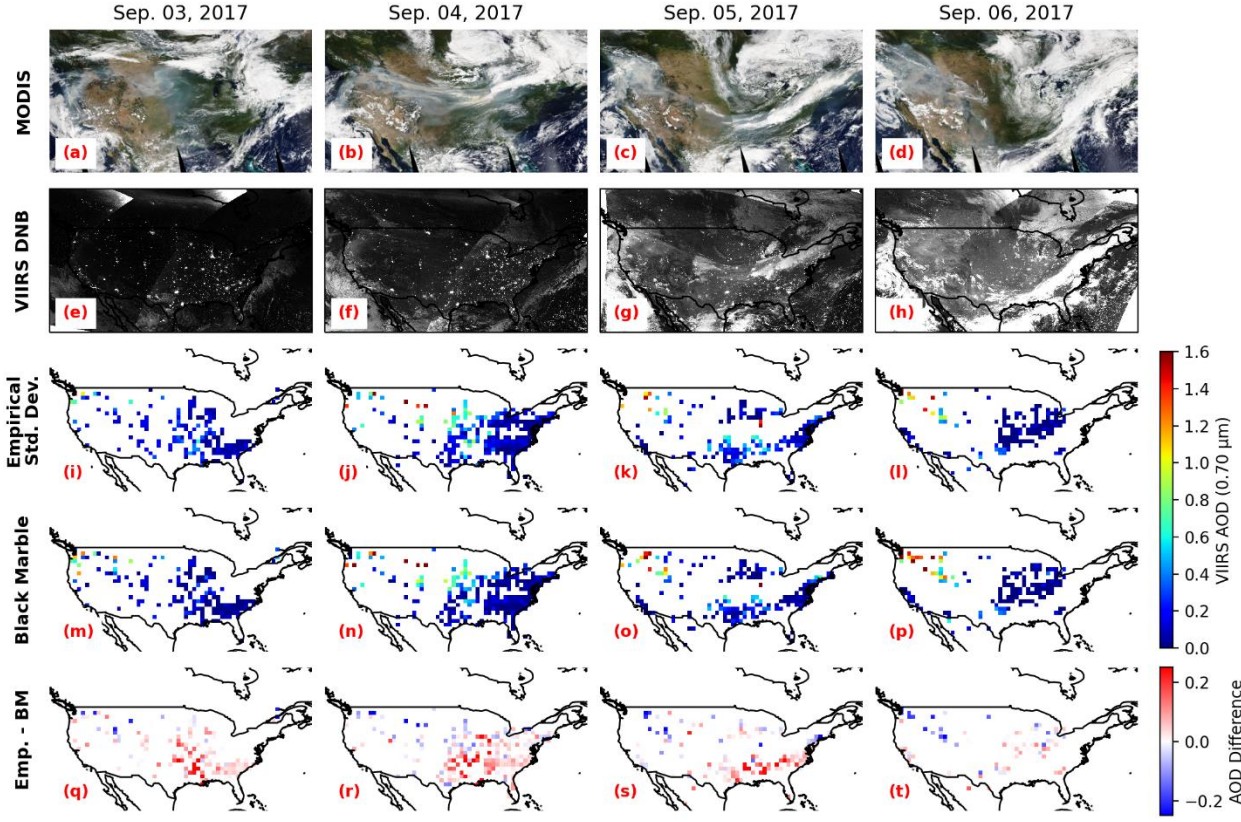

**Figure 10. a - d) Aqua MODIS true color images over USA for Sep. 03- Sep. 06, 2017. The MODIS true color images were obtained from the NASA Worldview website (https://worldview.earthdata.nasa.gov). e – h) Suomi-NPP VIIRS/DNB imagery over the USA for Sep. 03 – Sep. 06, 2017. i - l) Similar to e-h) but for VIIRS/DNB nighttime AOT retrievals derived using the empirically-based STD method for estimating $\Delta I_a$ values. m - p) Similar to i - l) but for VIIRS/DNB nighttime AOT retrievals derived using the NASA VNP46 data for estimating $\Delta I_a$ values. q – t) Similar to m – p), but showing the differences between the nighttime AOT from the empirically-based STD method and the VIIRS/DNB nighttime AOT retrievals.**

Figures 11a - 11d show the seasonally-averaged MISR AOT (550 nm) for the Indian Subcontinent region, while Figures 11e - 11h represent the seasonally-averaged Aqua MODIS DT AOT (550 nm) for the same region; Figures 11i-11l show similar seasonal averages of Aqua MODIS DB AOT (550 nm). Figure 11m - 11p show the seasonally-averaged nighttime VIIRS/DNB AOT for the same region using the empirically-based STD Method. Figure 11q - 11t are similar, but based on nighttime VIIRS/DNB AOT retrievals from the empirically-based Mean Method. In both approaches, for the DJF season, daytime MISR/MODIS AOT data suggested the region was in a relatively low aerosol loading scenario with higher AOT values occurring over northeastern India. For the MAM region, heavier aerosol plumes occurred over eastern India. For the JJA season, aerosol plumes occurred over northern India, just to the south of the Himalayas. Aerosol plumes occurred across the northern and central regions of India for the SON season. The AOT patterns, as indicated from MISR/MODIS, were mostly or partially captured by nighttime VIIRS/DNB AOT, as shown in Figure 11m – 11t.









**Figure 11. a - d) Seasonally averaged daytime MISR AOT (550nm) over the the Indian Subcontinent region for the DJF, MAM, JJA, and SON seasons. e – h) Similar to a – d), but for MODIS Dark Target (DT) AOD (550 nm). i - l) Similar to a - d), but for MODIS Deep Blue (DB) AOD (550 nm). m-p) Similar to a - d) but for VIIRS/DNB nighttime AOT retrievals derived using the empirically-based STD Method for estimating $\Delta I_a$ values.  q - t) Similar to m) - p) but using the empirically-based Mean Method for estimating $\Delta I_a$ values.  u – x) Similar to q – t), but showing the differences between the nighttime AOT from the empirically-based STD method and the empirically-based Mean method.**

Nighttime AOT patterns can also be observed on a daily basis in regions with heavier aerosol loading, such as the Indian Subcontinent. Figures 12a – d show Suomi-NPP VIIRS true color imagery from 6 – 9 November 2017, obtained from the NASA Worldview website (https://worldview.earthdata.nasa.gov/), with plumes of dense aerosol pollution across the northern portions of the Indian Subcontinent and against the Himalayas. The nighttime VIIRS DNB data for the same dates (shown in Figure 12 e – h) exhibit similar patterns, with aerosol plumes visible across the northern Indian Subcontinent at night. The nighttime VIIRS/DNB AOT derived using the STD-based empirical method (shown in Figure 12 i – l) match the patterns shown in both the VIIRS daytime true color and nighttime DNB imagery. Higher nighttime AOT values were found in northern India, with relatively lower AOT values found in the southern India Subcontinent.

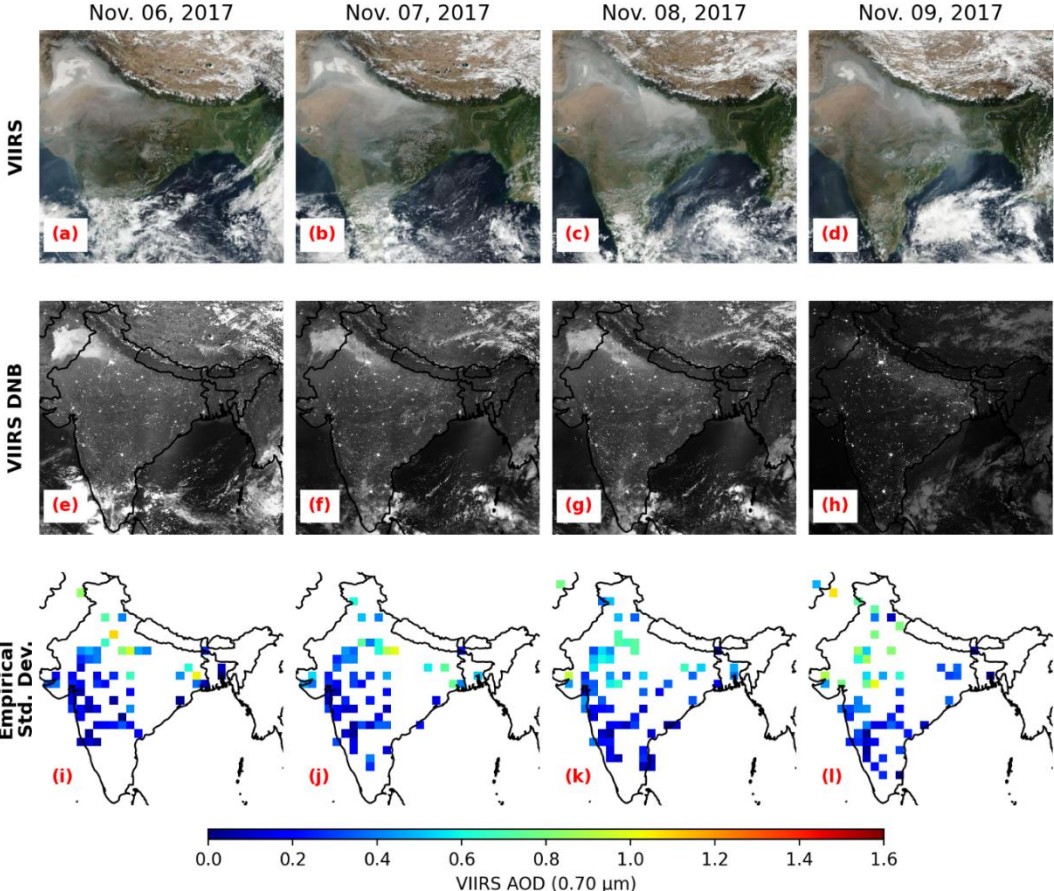

**Figure 12. a - d) Suomi-NPP VIIRS true color images over the Indian Subcontinent for Nov. 06 - Nov. 09, 2017, obtained from the NASA Worldview website (https://worldview.earthdata.nasa.gov). e – h) Suomi-NPP VIIRS/DNB imagery over the Indian SC for**



**Nov. 06 – Nov. 09, 2017. i - l) Similar to e - h) but for VIIRS/DNB nighttime AOT retrievals derived using the empirically-based**
**STD method for estimating $\Delta I_a$ values.**

Figures 13a - 13d show seasonally-averaged MISR AOT (550 nm) for the Middle East region. Figures 13e - 13h show the seasonally-averaged MODIS AOT (550 nm) for the same region, with Figures 13i - 13l having similar seasonal averages of Aqua MODIS DB AOT (550 nm). Relatively high aerosol loadings were found for the MAM and JJA seasons with lower aerosol loadings for the DJF and SON seasons of 2017. Lower aerosol loadings were also found for nighttime VIIRS/DNB
AOT retrievals using the empirically-based STD Method (Figures 13m -13p) and for the empirical-based Mean Method (Figures 13q - 13t) for the DJF and SON seasons for the Middle East region. Significant aerosol plumes observed for the JJA season from MODIS/MISR AOT retrievals are not as prominent from the nighttime VIIRS/DNB aerosol retrievals. A significant portion of the heavily polluted region, as shown in Figure 13c/13g, did not have VIIRS nighttime AOT data due to a paucity of artificial lights, which illustrates one of the limitations of nighttime aerosol retrieval using artificial light
sources.

As with the US and Indian Subcontinent domains, daily nighttime AOT patterns can be observed in the Middle East domain. Figures 14 a – d show Suomi-NPP VIIRS true color imagery, taken from the NASA Worldview website (https://worldview.earthdata.nasa.gov/), over the Middle East for 28 October to 31 October 2017. As shown in the true color imagery, a large dust plume moved southeast across Iraq and northern Saudi Arabia, eventually extending out over the
Persian Gulf on 31 October 2017. Both the nighttime VIIRS DNB imagery (Figures 14 e – h) and nighttime VIIRS/DNB AOT derived using the STD-based method (Figures 14 i – l) show similar patterns overnight. Very high nighttime AOT values greater than 1.0 were reported in northern Saudi Arabia on 29 October 2017, matching the location of the dust plume in the daytime VIIRS true color imagery. The nighttime AOT data also captured the plume as it crossed over Kuwait and into the northern Persian Gulf on 31 October 2017.

We also checked the nighttime VIIRS/DNB AOT production rates, as listed in Table 1. On average, at the 1° x 1° (Latitude/Longitude) grid cell size, ~40% of the 1cells had valid nighttime retrievals for the USA and Middle East regions, and ~30% of the cells had valid retrievals for the Indian Subcontinent region. For the cells with valid retrievals, over the entire year of 2017, an average of ~120 nights, ~145 nights and ~115 nights were found for the USA, Middle East, and Indian Subcontinent regions, respectively. Only slight differences were found for the retrieval production rates using the
different retrieval methods.




**Table 1. The total number of 1°x1° Lat/Lon grid for a given study region (US, Middle-East, the Indian Subcontinent), as well as the number of 1°x1° Lat/Lon grids with valid VIIRS nighttime AOT retrievals. For a grid with valid retrievals, the average number of nights with AOT retrievals is also reported.**

| Method | Region | Total grids (1x1° Lat/Lon) | Grids with valid AOT retrievals | Average number of nights of retrievals for a valid grid |
|---|---|---|---|---|
| Empirical-based Mean | US | 1051 | 405 | 120.0 |
| | ME | 1053 | 416 | 144.8 |
| | Indian SC | 461 | 137 | 116.5 |
| Empirical-based Median | US | 1051 | 405 | 120.1 |
| | ME | 1053 | 414 | 145.8 |
| | Indian SC | 461 | 137 | 116.3 |
| Empirical-based STD | US | 1051 | 404 | 120.6 |
| | ME | 1053 | 410 | 146.7 |
| | Indian SC | 461 | 139 | 115.1 |
| VNP46-based STD | US | 1051 | 404 | 120.6 |


**Figure 13. a - d) Seasonally averaged daytime MISR AOT (550nm) over the Middle East region for the DJF, MAM, JJA, and SON seasons. e – h) Similar to a – d), but for MODIS Dark Target (DT) AOD (550 nm). i - l) Similar to e - h), but for MODIS Deep Blue (DB) AOD (550 nm). m - p) Similar to a - d) but for VIIRS nighttime AOT derived using the empirically-based STD Method for estimating $\Delta I_a$ values. q - t) Similar to m - p) but using the empirically-based Mean Method for estimating $\Delta I_a$ values. u – x) Similar to q – t), but showing the differences between the nighttime AOT from the empirically-based STD method and the empirically-based Mean method.**

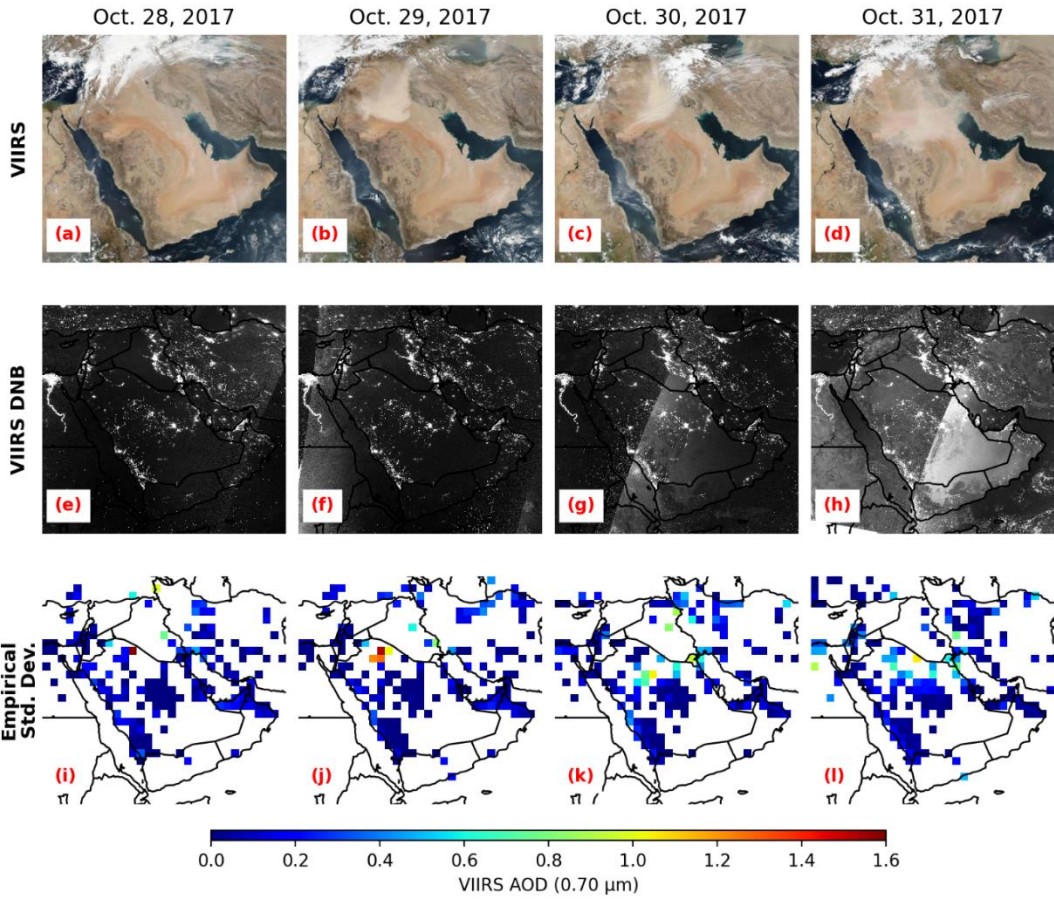

**Figure 14. a - d). Suomi-NPP VIIRS true color images over the Middle East for Oct. 28 - Oct. 31, 2017, obtained from the NASA Worldview website (https://worldview.earthdata.nasa.gov). e – h) Suomi-NPP VIIRS/DNB imagery over the Middle East for Oct. 28 – Oct. 31, 2017. i - l) Similar to e - h) but for VIIRS/DNB nighttime AOT retrievals derived using the empirically-based STD method for estimating $\Delta I_a$ values.**



## 4.5 Limitations and possible improvements

While these preliminary nighttime AOT results are promising, there are non-trivial limitations to the use of artificial light sources which must be acknowledged.

Firstly, artificial light sources may vary with time (e.g. Solbrig et al., 2020). Artificial lights from some cities may experience non-trivial changes depending on the hour, day, or season. To compensate for this temporal invariance issue, various QA steps (as noted in Sect. 3.2) need to be implemented in order to exclude cities/towns with unstable light sources throughout a period.

Second, due to the lack of visible channel data with low signal-to-noise ratios at new moon (and possibly waning crescent and waxing crescent) nights and reliance on infrared bands for cloud screening at night, cloud contamination remains problematic for the study. Also note that AERONET data used in the study are cloud screening already, thus the AERONET and VIIRS AOT comparison study may be cloud-free sky biased. Additional QA steps, such as testing the variability of city-center light sources, as well as the use of the radiance-to-standard-deviation relationship, can further be applied to cloud clearing the nighttime AOT retrievals.

Also, as suggested by serval studies (e.g., Solbrig et al., 2020; Kyba et al., 2022), city light variability is a function of viewing geometry. Although simple viewing angle correction, based on Zhang et al., (2019) was implemented in this study, it is anticipated that multi-angle nighttime observations are needed for carefully quantifying angular dependence of artificial nighttime lights for various applications, including nighttime aerosol retrievals.

Lastly, we found that $\Delta I_a$ / $dI_a$ values are critical for nighttime aerosol retrievals. However, for the empirically-based Methods discussed in this study, one $\Delta I_a$ / $dI_a$ value per city was derived for a year, and this ignores seasonal variations in $\Delta I_a$ / $dI_a$ values. The NASA VNP46 data can be used to estimate $\Delta I_a$ values on the daily basis. However, a significant low bias exists when using the NASA VNP46 data to estimate $\Delta I_a$ values over heavily polluted regions. Thus, we anticipate improvements in the NASA VNP46 data for it to be used in future nighttime aerosol retrievals.

The QA steps used in the nighttime aerosol retrieval also have a side effect of at times screening out high AOT aerosol events which are of particular interest for the users of such observations including the aerosol prediction community. It should be noted that this problem does not solely occur with the nighttime retrievals presented here, but is also a known issue with daytime AOT retrievals associated with cloud screening. Further, the nighttime retrievals were found to have a low AOT bias under certain conditions. In data assimilation space, both can lead to low biases in analysis AOT and in subsequent forecasts. Adjustments to the QA steps may be tested in the future to help prevent screening of high aerosol loading conditions. Additionally, bias correction can be applied to the product prior to use in data assimilation, as has been done for daytime AOT retrievals (Zhang and Reid, 2006; Hyer et al. 2011) in order to maximize the utility of the nighttime aerosol product.





# 5    Conclusions and Implications

In this study, we leveraged 2017 VIIRS/DNB data to explore the feasibility of nighttime aerosol retrievals over the USA, the Indian Subcontinent, and the Middle East. Three key questions were addressed: 1) Can the retrievals be performed on an equal-area grid?; 2) To what extent can NASA's Black Marble data be used to estimate aerosol-free sky artificial light emissions ($\Delta I_a$ values) for nighttime retrievals?; and 3) Do there exist alternate methods for estimating the spatial derivative of radiances (other than standard deviation method) that are capable of providing robust nighttime AOT retrieval results? Our findings demonstrate that:

1)  Nighttime Aerosol Optical Thickness (AOT) can indeed be effectively retrieved on equal-area grids, reducing the need for prior knowledge of urban areas and simplifying the computational process.

2)  NASA Black Marble data are suitable for estimating $\Delta I_a$ values in relatively clean regions such as the USA. However, in highly polluted areas like the Indian Subcontinent, significant low biases in AOT were observed, indicating the potential contamination of Black Marble data by semi-persistent elevated aerosol loading. Future integration of nighttime AOT retrievals to the NASA Black Marble retrievals, as presented in this study, may help reduce these biases.

3)  The spatial derivative of radiances from artificial light sources, whether estimated by the empirically-based STD, Mean, or Median Methods, yields comparable nighttime AOT retrieval results, suggesting that multiple approaches could be applied in ensemble methods to potentially enhance retrieval accuracy.

While the current study bears conceptual similarity to reverse AERONET methods, where cities act as emission sources and the VIIRS/DNB as the sensor, several caveats need addressing. Artificial light sources can vary non-trivially over time, and also, cloud contamination remains a challenge. Screening for stable artificial light sources and innovative cloud-clearing methods are necessary for further operational retrieval efforts. Despite these limitations, the results of this study open new avenues for the operational use of artificial light sources for nighttime aerosol retrievals. The methods developed could be deployed using VIIRS DNB aboard the current and upcoming JPSS satellites to complement the daytime retrievals currently done operationally.

Such nighttime aerosol products as demonstrated in this work, with the addition of appropriate up-front cloud screening, could be very impactful for numerical aerosol prediction systems that rely on satellite observations for model analyses via data assimilation. As most of the current operational aerosol prediction systems rely on the assimilation of daytime aerosol optical thickness retrievals (Zhang et al. 2008; Benedetti et al. 2009; Lynch et al. 2016; Rubin et al. 2016; Xian et al. 2019), incorporation of nighttime AOT products with associated uncertainty estimates would be straightforward and should be impactful in capturing the diurnal variability of aerosol. As the methods demonstrated in this work require a sufficient geographic distribution of city lights, the method could be expanded to other populated regions around the globe for an expanded and improved nighttime AOT product.

In conclusion, this research paves the way for future operational systems capable of enabling continuous/diurnal aerosol monitoring, offering new information that is crucial not only for regions directly prone to elevated nighttime AOT events and associated hazardous air quality, but also to locations downstream of these production areas. Given the enduring operation of VIIRS/DNB as part of a long-term operational program, and with the anticipated proliferation of low-light visible technology on future sensors on international programs, the current work provides a foundation for global aerosol retrievals at night that aims to augment the climate data record. The fundamental importance of such work lies in its potential to expand the scope of satellite-based aerosol observations, offering more comprehensive insights into aerosol distributions across the diurnal cycle, especially in regions where nighttime environmental properties and have been historically under-monitored.

**Code and data availability:**

The Aqua MODIS Deep Blue and Dark Target aerosol data as well as the NASA Black Marble data products (VNP46) were obtained from the NASA Level-1 and Atmosphere Archive & Distribution System Distributed Active Archive Center (LAADS DAAC; https://ladsweb.modaps.eosdis.nasa.gov/).   The true color MODIS images were obtained from the NASA world view website (https://worldview.earthdata.nasa.gov).  The VIIRS data were obtained from the NOAA Comprehensive Large Array-Data Stewardship System (CLASS) site (https://www.aev.class.noaa.gov/saa/products/welcome).  The NASA AERONET data were obtained from the NASA AERONET website (https://aeronet.gsfc.nasa.gov/).

**Author contributions**

J. Zhang and J. Reid designed the study.  B. Sorenson and S. Jaker assisted with data and/or image processing.  S. Miller provided the nighttime lunar model.  M. Román and Z. Wang assisted with the NASA Black Marble data products analyses. All authors participated in drafting the manuscript and/or providing critical comments for the study.

**Competing interests**

T. Eck is a member of the editorial board of Atmospheric Measurement Techniques.

**Financial support:**

This project is supported by the NASA grant 80NSSC20K1748.



**Acknowledgments:**

We thank the NASA AERONET team for the surfaced-based aerosol optical thickness data. Support of the NOAA JPSS
Program Office is gratefully acknowledged. We thank the NASA world view website for the true color Aqua MODIS/VIIRS
images.

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
