# Peer review of "Towards Gridded Nighttime Aerosol Optical Thickness Retrievals Using VIIRS Day/Night Band Observations Over Regions with Artificial Light Sources"

_Atmospheric Measurement Techniques, 2024_

## Referee Comment (RC1)

This manuscript demonstrates innovation and practical value in the research of nighttime aerosol optical thickness retrieval methods, with a clear description and adequate demonstration. The proposed method has practical implications for monitoring air quality and aerosol loading, especially in regions where nighttime data are crucial but traditionally lacking. However, there are still some contents and details that need further improvement and discussion.

1. The manuscript compares nighttime VIIRS/DNB AOT retrievals with other spatial collections of aerosol observations, such as daytime satellite AOT from MISR and MODIS. How significant is the impact of wavelength differences on the comparative analysis of AOT?

2. The author's analysis is comprehensive, but it is recommended to further delve into the anomalies and biases present in the results, and their potential implications on the accuracy of AOT retrieval. Furthermore, the author may consider incorporating additional discussions on the impact of aerosol types, seasonal variations, and geographical distributions on the results, in order to offer a more profound understanding.

3. In Figure 8, the blue and black scatter points are not easily distinguishable. Please use more contrasting colors or apply a color gradient to differentiate the frequency of the data points.

4. The manuscript mentions the portability and universality of the algorithm as one of the strengths of this study. The author is advised to further discuss the specific application scope and limiting conditions of this portability and universality.

---

## Author Comment (AC1)

*This manuscript demonstrates innovation and practical value in the research of nighttime aerosol optical thickness retrieval methods, with a clear description and adequate demonstration. The proposed method has practical implications for monitoring air quality and aerosol loading, especially in regions where nighttime data are crucial but traditionally lacking. However, there are still some contents and details that need further improvement and discussion.*

Response: We thank the reviewer for the constructive comments.

*1. The manuscript compares nighttime VIIRS/DNB AOT retrievals with other spatial collections of aerosol observations, such as daytime satellite AOT from MISR and MODIS. How significant is the impact of wavelength differences on the comparative analysis of AOT?*

Response: MODIS and MISR AOTs are available at 550 nm. The VIIRS DNB AOTs are available at 700 nm. There are non-negligible differences in AOTs from 550 nm to 700 nm. Still, we used MODIS and MISR data here simply to show the spatial patterns of aerosol plumes and inter-compare with spatial patterns of aerosol plumes from VIIRS DNB. No intent is made to compare absolute AOT values directly, as converting AOTs from 550 nm to 700 nm spectral channels is not an easy task (can be a study of its own) and thus is not included in the study.

We added the following discussions in the paper:

"Note that MODIS and MISR AOTs are available at 550 nm. The VIIRS DNB AOTs are available at 700 nm. There are non-negligible differences in AOTs from 550 nm to 700 nm. Here, we used MODIS and MISR data simply to show the spatial patterns of aerosol plumes and inter-compare with spatial patterns of aerosol plumes from VIIRS DNB. No intent is made to compare absolute AOT values directly, as converting AOTs from 550 nm to 700 nm spectral channels is not an easy task (can be a study of its own) and thus is not included in the study."

*2. The author's analysis is comprehensive, but it is recommended to further delve into the anomalies and biases present in the results, and their potential implications on the accuracy of AOT retrieval. Furthermore, the author may consider incorporating additional discussions on the impact of aerosol types, seasonal variations, and geographical distributions on the results, in order to offer a more profound understanding.*

Responses: Uncertainties in AOT retrievals were extensively discussed in the paper, including a section (Section 4.5) that discussed limitations and potential improvements of the study. The impacts due to aerosol types were included in Section 4.3.2. Comparisons with daytime AOT retrievals were also conducted on both a seasonal and regional basis.

Still, we added absolute errors in VIIRS and lunar AERONET comparisons in the text. We also added the following discussions in the conclusion section.

"The uncertainties/biases in estimated nighttime AOTs from the study contributed mostly from uncertainties in estimated aerosol-free sky artificial light emissions or $\Delta I_a$ values, including angular dependence of $\Delta I_a$ values. Erroneous aerosol typing, cloud contamination, and the size of artificial light sources can also contribute to biases and anomalies in retrieved AOT values."

*3. In Figure 8, the blue and black scatter points are not easily distinguishable. Please use more contrasting colors or apply a color gradient to differentiate the frequency of the data points.*

Response: We revised Figure 8. We changed the blue color to light blue to enhance the contrast (note that the high quality .ps file for figure 8 is also available).

*4. The manuscript mentions the portability and universality of the algorithm as one of the strengths of this study. The author is advised to further discuss the specific application scope and limiting conditions of this portability and universality.*

Responses. In previous efforts, city locations are needed and retrievals were performed over selected cities. This requires a database of global cities. In this approach, a study region is divided in equal grids, and retrievals can be performed for each grid as long as there are sufficient artificial light pixels (e.g. > 50). This provides a flexibility for performing nighttime aerosol retrievals using artificial lights. We already had related discussions in the paper (discussed in the introduction section and Section 3.2). Still, to perform retrievals, aerosol and cloud free sky artificial light emissions are needed; this requires collection of data from a long period (one year in this study). Thus, temporal variation in artificial light sources, which could introduce uncertainties to the study, are not included. We also realized cloud contamination remains a problem for implementing the developed algorithm and similar discussions already included in Section 4.5. We believe we already discussions major issues we could realize in the text.

---

## Author Comment (AC2)

*This study utilizes VIIRS DNB day-night band data to investigate the spatial derivative of upward atmospheric layer attenuation and artificial light at night (ALAN) over the United States, the Middle East, and the Indian subcontinent in 2017. It explores the feasibility of developing a gridded nighttime aerosol optical thickness (AOT) dataset. Additionally, the study evaluates the potential of using the NASA standard Black Marble nighttime lights product suite (VNP46) to estimate the spatial derivative of surface artificial light emissions, and discusses the sensitivity of nighttime aerosol retrieval to observational conditions as well as the application of different methods for estimating the spatial derivative of surface artificial light emissions. The research validates the retrieved AOD by comparing it with ground-based AOD data from AERONET sites and satellite-based AOD products, such as MODIS AOD and MISR AOD. There are certain aspects/details that require further clarification from the authors, and I have listed them in my comments below :*

We thank the reviewer for the constructive comments.

*1. The selection of aerosol types has a certain impact on the inversion accuracy. In this context, aerosol plumes in the Indian subcontinent primarily consist of a mixture of polluted haze, smoke, and pollutant aerosols (Line 224). How is this mixed aerosol type (aerosol composition) defined?*

Response:  Great question.  Since there is only 1 channel (Day-Night-Band) available at nighttime, it is not possible to determine aerosol types from VIIRS DNB observations alone. Therefore, we simply preassigned aerosol type for each region.  We assigned pollutant aerosol to the Indian subcontinent, knowing there might be uncertainties in the aerosol typing assignment over the region.  Indeed, it is a region with a mixture of pollutant haze, smoke and pollutant aerosols.  The aerosol type assignment will affect diffuse radiance correction and the uncertainties in this misidentification of aerosol types is further discussed in section 4.3.2.

*2. The lunar AERONET AOT data at 440, 675, 870, 1020, and 1640 nm (Line 152).  Why was the 700 nm value not obtained by fitting the 675 nm and 870 nm bands for the analysis in Figures 4-7?*

Response: We chose to use AERONET data from 675 nm to inter-compare with VIIRS DNB retrievals from 700nm for two reasons.  First, only marginal changes are expected from AOTs from 675 and 700 nm spectral channels, due to the small spectral gap between the two channels. Second, uncertainty exists in interpolating 700 nm AERONET data using AERONET data from 675 and 870 nm.  Given the above two reasons, we used AERONET data from the 675 nm instead.

We added the following discussions in the text:

"Note that we chose to use AERONET data from 675 nm to inter-compare with VIIRS DNB retrievals from 700nm for two reasons.  First, only marginal changes are expected from AOTs

from 675 and 700 nm spectral channels, due to the small spectral gap between the two channels. Second, uncertainty exists in interpolating 700 nm AERONET data using AERONET data from 675 and 870 nm.".".

*3. Please discuss the influence of lunar radiation on the AOD nighttime inversion results in Section 4.5.*

Response: We discussed this topic in detail in one of our other papers, targeting to understanding nighttime AOT retrievals using a nighttime 3-D radiative transfer model [Zhang et al., 2023]. We added the following discussions in the text to address the comment.

"This finding is not a surprise as we have also explored the topic using a nighttime 3-D radiative transfer model [Zhang et al., 2023]. Zhang et al. [2023] suggests that incoming lunar flux introduces only marginal impacts on nighttime AOT retrieving using the artificial light based method."

*4. The 6S model requires satellite surface reflectance data as input, but the method for calculating the surface reflectance from VIIRS DNB remote sensing imagery is not explained.*

Response: We used the 6S model to estimate the ratio of direct versus diffused downward moon flux/radiation due to the atmospheric layer. Since only the atmospheric layer is involved in the process, the change in surface properties does not affect the ratio. Details of the approach are captured in our earlier papers [e.g. Johnson et al., 2013].

We added the following discussion in the text:

"Details of using the 6S model for estimating $k$ terms are included in a s previous paper (see Johnson et al., 2013)."

Johnson, R. S., Zhang, J., Hyer, E. J., Miller, S. D., and Reid, J. S.: Preliminary investigations toward nighttime aerosol optical depth retrievals from the VIIRS Day/Night Band, Atmos. Meas. Tech., 6, 1245–1255, https://doi.org/10.5194/amt-6-1245-2013, 2013.